# SyncTrack: Rhythmic Stability and Synchronization in Multi-Track Music Generation

**Hongrui Wang[1][*], Fan Zhang[1][*][†], Zhiyuan Yu[2], Ziya Zhou[3], Xi Chen[3], Can Yang[1,4][†], Yang Wang[5][†]**

[1]Department of Mathematics, The Hong Kong University of Science and Technology
[2]State Key Lab of CAD&CG, Zhejiang University
[3]Academy of Interdisciplinary Studies, The Hong Kong University of Science and Technology
[4]State Key Laboratory of Nervous System Disorders, The Hong Kong University of Science and Technology
[5]The University of Hong Kong
{hwangfb, zzhoucp, xchengx}@connect.ust.hk, yang.wang@hku.hk
{macyang, mafzhang}@ust.hk, zhiyuan.yu@zju.edu.cn
https://synctrack-v1.github.io

## Abstract

Multi-track music generation has garnered significant research interest due to its precise mixing and remixing capabilities. However, existing models often overlook essential attributes such as rhythmic stability and synchronization, leading to a focus on differences between tracks rather than their inherent properties. In this paper, we introduce SyncTrack, a synchronous multi-track waveform music generation model designed to capture the unique characteristics of multi-track music. SyncTrack features a novel architecture that includes track-shared modules to establish a common rhythm across all tracks and track-specific modules to accommodate diverse timbres and pitch ranges. Each track-shared module employs two cross-track attention mechanisms to synchronize rhythmic information, while each track-specific module utilizes learnable instrument priors to better represent timbre and other unique features. Additionally, we enhance the evaluation of multi-track music quality by introducing rhythmic consistency through three novel metrics: Inner-track Rhythmic Stability (IRS), Cross-track Beat Synchronization (CBS), and Cross-track Beat Dispersion (CBD). Experiments demonstrate that SyncTrack significantly improves the multi-track music quality by enhancing rhythmic consistency.

## 1 Introduction

Advances in music generation from raw audio have enabled systems to produce music conditioned on user specifications such as genre, mood, tempo, and instrumentation. The mixed waveforms produced by current music generation methods often lack the flexibility needed for professional editing, limiting their usefulness for musicians (Cano et al., 2018). In music production, it's important to manipulate individual instrument tracks, including mixing, rearranging or adding new instruments. Recently, many works (Mariani et al., 2023; Karchkhadze et al., 2025; Parker et al., 2024; Yao et al., 2025b) focus on multi-track audio generation. Unfortunately, beat stability and synchronization are not considered in generation of multi-track music and its quality evaluation. As the primary organizing element of time in music, rhythm provides the essential framework upon which melody and harmony are built (Abrassart & Doras, 2022). Temporal prediction errors caused by irregular beats or inter-track desynchronization lead to persistent violations of auditory expectations, inducing listener discomfort (Mas-Herrero et al., 2018).

To date, existing methods are incapable of generating satisfactory music with coherent and stable rhythm due to the mismatch between their model architecture and the inherent properties of multi-track

---

[*]Equal contribution.
[†]Corresponding author.

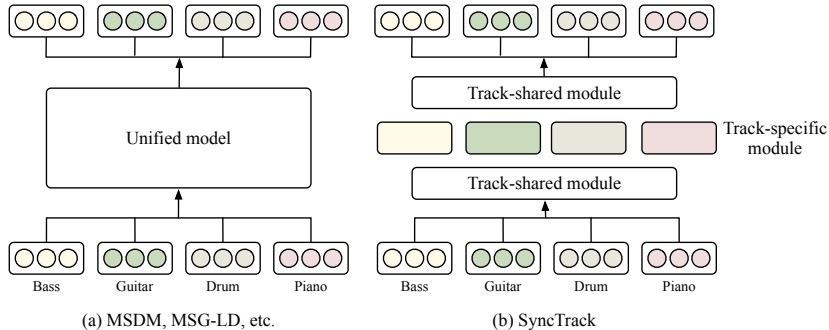

Figure 1: (a) Previous methods (Mariani et al., 2023; Karchkhadze et al., 2025) leverage a unified model to learn the joint distribution of multi-track audio stems. (b) While our proposed SyncTrack incorporates both track-shared modules and track-specific modules for common and specific information between tracks.

music. As shown in Fig. 1(a), MSDM (Mariani et al., 2023) and MSG-LD (Karchkhadze et al., 2025) aim to learn the joint distribution of multi-track audio stems. They treat this task as multivariable time series (MTS) or video generation, which contains complex interactions and huge discrepancy between variables or channels (Wang et al., 2022; Qiu et al., 2025). In contrast to MTS and video generation, multi-track music generation faces unique challenges. In the multi-track music generation task, we need to simultaneously generate music for multiple tracks, with each track representing independent layers or channels in a musical piece, for example, bass, drum, and so on. Two core aspects govern this process: 1) Rhythm refers to the structured temporal arrangement of musical events. Within a single track, rhythm must exhibit stability, meaning that note onsets and beats adhere consistently to a regular metrical grid. Across multi tracks, rhythm requires synchronization, ensuring that events from different instruments align precisely with each other and a shared underlying pulse. 2) Timbre defines the unique perceptual quality of sound that distinguishes one instrument from another, such as the warm character of a bass or the bright attack of a piano. These two types of information are different: 1) good musical rhythms possess synchrony and are more harmonious, so rhythm information is shared across tracks; 2) timbre, on the other hand, is the unique information retained by each track, with tracks being mutually independent. MSDM and MSG-LD overemphasize on the inter-track discrepancies, overlooking the common rhythm information. Consequently, neither model achieves satisfactory results in either subjective audio quality or rhythmic consistency.

To solve the above challenges, we propose a synchronous multi-track waveform music generation model, named SyncTrack, with a novel architecture suitable for capturing data characteristics. Specifically, to handle common and specific information of multiple tracks, SyncTrack adopts a unified architecture that incorporates both track-shared modules and track-specific modules, as shown in Fig. 1(b). In the track-shared modules, we devise two types of cross-track attention submodules to further capture cross-track rhythmic stability and synchronization: 1) *global cross-track attention* is essential for modeling global stability, ensuring that all tracks maintain a consistent rhythm framework over the entire piece; 2) *time-specific cross-track attention* is critical for fine-grained synchronization, aligning musical events across tracks at the same temporal position. In the track-specific modules, we construct the learnable instrument prior for each track. We incorporate the prior into the latent representation of each track in the track-specific modules. By doing so, timbre and other track-specific features can better capture the differences between tracks. Furthermore, the separation between track-shared and track-specific modules is structurally simple and could be dropped into other latent-audio diffusion systems.

Furthermore, we innovatively propose introducing rhythmic consistency in the evaluation of multi-track music quality. Current works assess the quality of multi-track audio generation merely using the Fréchet Audio Distance (FAD) (Kilgour et al., 2018), which cannot quantify the stability and synchronization. FAD measures the similarity between generated and reference audio samples by calculating their distributional distance. However, the sequential yet highly structured nature of audio data presents challenges for accurate FAD computation (Yang & Lerch, 2020). Hence, FAD compresses the audio file into VGGish embeddings (Kilgour et al., 2018), which overly compress temporal information and limit the ability to access stability and synchronization. In this paper, we

propose three novel metrics, Inner-track Rhythmic Stability (IRS), Cross-track Beat Synchronization (CBS) and Cross-track Beat Dispersion (CBD), to evaluate these properties. Specifically, **IRS** evaluates the rhythmic stability of an individual audio track based on the variance of its beat intervals. **CBS** quantifies the proportion of rhythmically aligned beats by employing a sliding tolerance window method. **CBD** computes the timing errors between aligned beats, providing a more refined measurement of beat synchronization. The combination of these three metrics with FAD enables a more comprehensive and accurate evaluation of multi-track music generation quality.

In summary, our contributions are as follows: 1) We propose a synchronous multi-track waveform music generation model, named SyncTrack. By incorporating both track-shared modules and track-specific modules, SyncTrack can effectively handle common and track-specific information; 2) We design two types of cross-track attention submodules, *global cross-track attention* and *time-specific cross-track attention* for global stability and fine-grained synchronization, respectively; 3) We innovatively propose the introduction of rhythmic consistency in the evaluation of multi-track music quality. Three metrics, **IRS**, **CBS** and **CBD** quantitatively evaluate rhythmic stability of an individual audio track and rhythmic synchronization alignment cross tracks; 4) Empirical results confirm that SyncTrack achieved better performance of multi-track music generation than existing state-of-the-art methods.

## 2 RELATED WORK

### 2.1 RHYTHMIC STABILITY AND SYNCHRONIZATION

Musical rhythm, as a core element of musical expression, has long been a focal point in both musicology and cognitive science (Mirka, 2004). Rhythmic stability and synchronization serve as the temporal backbone of musical organization and the foundation for effective collaboration in multi-track music composition (Abrassart & Doras, 2022; Hennig, 2014). Rhythmic stability and synchronization are sensitive to listeners; even untrained listeners are highly sensitive to temporal fluctuations or deviations in music (Large & Palmer, 2002). Temporal prediction errors caused by irregular beats or inter-track desynchronization lead to persistent violations of auditory expectations, inducing listener discomfort (Mas-Herrero et al., 2018). Consequently, rhythmic inconsistencies can disrupt auditory expectations and degrade the overall musical experience, making the maintenance of coherent rhythm a core challenge in computational music research.

### 2.2 AUDIO GENERATION AND MULTI-TRACK WAVEFORM MUSIC GENERATION

**Audio Generation**. Audio generation has rapidly progressed from early autoregressive models like WaveNet (Van Den Oord et al., 2016) and SampleRNN (Mehri et al., 2016), which generate audio sample by sample, to higher-level discrete token models and advanced generative approaches. Vector quantization method, such as VQ-VAE (Razavi et al., 2019), encode audio into compact code sequences, enabling models like Jukebox and MusicGen to capture long-range dependencies and facilitate efficient decoding. Recent advances utilize diffusion models—such as DiffWave (Kong et al., 2020b), WaveGrad (Chen et al., 2020), and latent diffusion frameworks like AudioLDM (Liu et al., 2023) and AudioLDM2 (Liu et al., 2024)-to further improve audio fidelity, scalability and controllability. These techniques have since been adapted for targeting musical domain like Riffusion (Forsgren & Martiros, 2022), Moûsai (Schneider et al., 2024), MusicLDM (Chen et al., 2024), and Stable Audio (Evans et al., 2024a; 2025; 2024b).

**Multi-track Waveform Music Generation.** Despite these advances, most models produce a single mixed waveform without access to individual instrument tracks. This limitation hinders downstream applications for professional musicians, such as remixing, adaptive arrangement, and track-wise editing (Cano et al., 2018). Multi-track music generation aims to address this gap by generating separate stems for different musical elements. Recent approaches treat multi-track music as structured, interdependent components: for instance, StemGen (Parker et al., 2024) and Jen-1 Composer (Yao et al., 2025b) use transformers and latent diffusion models to generate multiple canonical stems conditioned on prompts. Other methods, such as Multi-Source Diffusion Models (MSDM) (Mariani et al., 2023) and MSG-LD (Karchkhadze et al., 2025), extend diffusion models to jointly model fixed or variable sets of tracks within unified frameworks.

However, current model architectures are incompatible with the inherent properties of multi-track music, as they do not separately process track-specific and common information. As a result, models tend to focus on inter-track differences while neglecting common rhythmic patterns.

## 2.3 Synchronized Generation in Audio Related Domains

**Video-to-audio Generation.** This task seek to generate audio from video content, aiming to achieve both semantic consistency and temporal synchronization. Recent methods for synchronized video-to-audio generation primarily model alignment through rhythmic and hierarchical visual conditioning. Models like Diff-BGM (Li et al., 2024b) and LORIS (Yu et al., 2023) explicitly leverage dynamic motion features to control the rhythm and timing of the generated audio, ensuring beats align with on-screen action. Another trend involves disentangling visual semantics from rhythm using hierarchical features. For instance, VidMusician (Li et al., 2024a) employs global features for semantic style and local features for rhythmic cues, while VidMuse (Tian et al., 2025) utilizes long-short-term modeling. An alternative approach, Mel-QCD (Wang et al., 2025), deconstructs audio into components that are more predictable from video, enabling precise synchronization control for a pre-trained generator.

In this field, the objective evaluation on generated audio is based on audio quality, cross-modal alignment and rhythm alignment (Li et al., 2024a). To investigate rhythm alignment (referred to as temporal synchronization), some works (Yu et al., 2023; Li et al., 2024a) use beats coverage scores (BCS), beats hit scores (BHS) for evaluation. These metrics usually estimate beats from audio at a coarse, second-wise granularity. While they are suitable for simple audio forms, they cannot capture the precise rhythmic interactions necessary for multi-track music generation.

**Accompaniment Generation**. In this dmoain, models like SingSong (Donahue et al., 2023) enforce rhythmic alignment with input vocals; however, evaluations often rely on subjective listening tests rather than objective metrics. Such subjective evaluations require substantial human labor and time; thus, they are difficult to use for rapid and consistent model comparison during iterative development.

## 2.4 Objective Evaluation Metrics for Multi-Track Music Generation

In the area of **multi-track music generation**, objective evaluation metrics remain limited. The most commonly used metric for waveform music generation is the Frechet Audio Distance (FAD) (Kilgour et al., 2018), which measures the distributional distance between generated and reference audio samples. Although FAD is effective in capturing overall audio quality and similarity, it falls short in reflecting important musical characteristics such as rhythmic stability and cross-track synchronization. This limitation stems from the fact that FAD relies on compressed embeddings extracted by VG-Gish (Kilgour et al., 2018), which significantly reduce temporal resolution. The sequential and highly structured nature of music makes accurate FAD computation challenging (Yang & Lerch, 2020), as the temporal and rhythmic information crucial for music perception is lost during embedding compression. KAD (Chung et al., 2025) improves on FAD by using a distribution-free, unbiased estimator, offering a more efficient and perceptually aligned metric. SongEval (Yao et al., 2025a) addresses the limitations of objective metrics by incorporating multidimensional ratings from professional musicians, capturing broader aspects of musical perception. However, these newly proposed metrics still fail to adequately capture fine-grained rhythmic issues for multi-track music.

In the domain of **symbolic music generation**, there have been attempts to evaluate alignment using specialized metrics (Yu et al., 2022; Dong et al., 2018). However, these metrics are tailored for symbolic representations and cannot be directly applied to multi-track waveform audio generation due to differences in data modality and the complexity of raw audio signals.

Multi-track music demands fine-grained assessments that evaluate not only the rhythmic stability within each track but also the synchronization and interplay between different tracks. To address these gaps, we propose three novel metrics tailored for evaluating multi-track music generation: IRS, CBS and CBD. When combined with FAD, these complementary metrics provide a more comprehensive and musically meaningful assessment of generation quality, capturing both audio fidelity and the intricate rhythmic relationships critical for multi-track compositions.

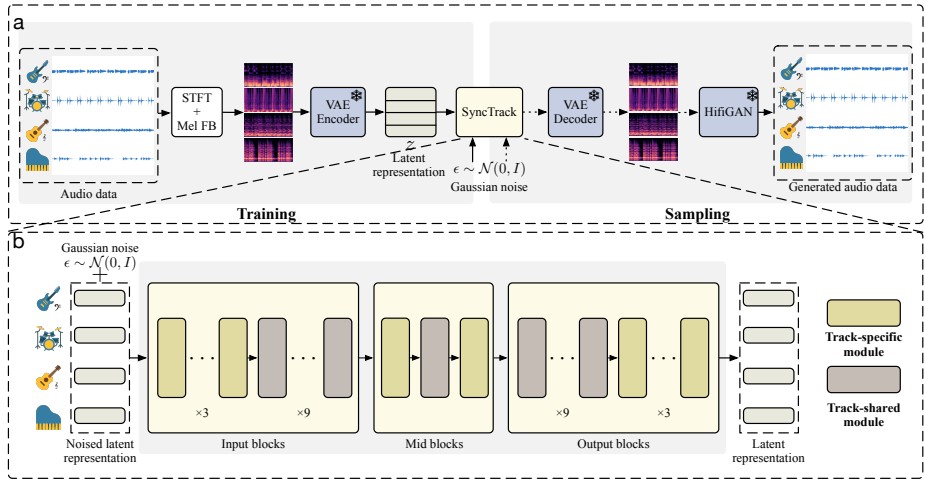

Figure 2: **a.** Overall pipeline for SyncTrack. *Training pipeline:* We train a four-track latent diffusion model. Each track is perturbed based on $l$-th signal-to-noise ratio. The model is optimized to predict the added noise $\epsilon \in \mathcal{N}(0, I)$. More details are in Section 3.1. *Inference pipeline:* At test time, four-track latents are generated and then decoded into audio data. **b.** SyncTrack consists of input, mid, and output blocks, which contains track-specific modules and track-shared modules.

## 3 SYNCTRACK: SYNCHRONOUS MULTI-TRACK MUSIC GENERATION MODEL

In this work, we propose a novel multi-track music generation model based on the Latent Diffusion Model (LDM) framework (Rombach et al., 2022; Ho et al., 2020). By learning the underlying probability distribution of the given dataset, SyncTrack can generate new multi-track music samples approximately from the same underlying distribution.

**Overall Framework.** As shown in Fig. 2a, audio data $\{x^s\}_{s=1}^S \in \mathbb{R}^{T'}$ with length $T'$ is first encoded into latent representation in two steps: 1) audio data to mel-spectrogram using Short-Time Fourier Transform (STFT) and a Mel filter bank; 2) mel-spectrogram to latent representation using a pre-trained Variational Autoencoder (VAE) (Kingma & Welling, 2013), which is formulated as follows:

$$z^s := \text{VAE}_{\text{enc}}(\text{STFT\&MelFB}(x^s)) \in \mathbb{R}^{C \times T \times F}, \tag{1}$$

where $T$ and $F$ are the temporal and frequency dimensions after compression, and $C$ represents the hidden dimension.

SyncTrack aims to learn the distribution of $z^s$, which is difficult to transform from an easy-to-sample distribution. Inspired by DDPM (Ho et al., 2020), we set a sequence of data distributions perturbed by $L$ signal-to-noise ratio levels. SyncTrack iteratively improves the signal-to-noise ratio of the perturbed data distribution from $l$ to $l - 1$, ultimately generating multi-track music when signal-to-noise ratio reaches 1. The way to improve the signal-to-noise ratio is to predict the added noise on the perturbed data. Specifically, given the latent representation $\{z^s\}_{s=1}^S$, we obtain the perturbed data $\{z_l^s\}_{s=1}^S$ by adding noise $\epsilon$ to $\{z^s\}_{s=1}^S$ at the $l$-th signal-to-noise ratio. SyncTrack $\epsilon_\theta$ learns to predict the added noise:

$$\mathcal{L}(\theta) = \mathbb{E}_{\epsilon \sim \mathcal{N}(0,I), \{z^s\}_{s=1}^S, l} \left\| \epsilon - \epsilon_\theta(\{z_l^s\}_{s=1}^S, l) \right\|^2. \tag{2}$$

where $l$ is uniformly sampled from $\{1, \cdots, L\}$.

In the sampling phase, as shown in Fig 2a, we can utilize the trained SyncTrack to approximate the distribution of $z^s$, Then the sampled $\hat{z}^s$ is decoded into audio data of $s$-th track by leveraging the VAE decoder and HiFi-GAN vocoder (Kong et al., 2020a):

$$\hat{x}^s = \text{HiFiGAN}\left(\text{VAE}_{\text{dec}}(\hat{z}^s)\right). \tag{3}$$

As shown in Fig. 2b, for track-specific and common information between tracks, SyncTrack is designed as a unified architecture incorporating both track-shared modules and track-specific modules. Next we delve into details of these two types of modules.

**Track-shared modules.** As shown in Fig. 3a, each track-shared module consists of the ResBlock (He et al., 2016), inner-track attention, global cross-track attention and time-specific cross-track attention. Note that the inner-track attention retains the same architecture as that commonly used in the 2D U-Net (Ronneberger et al., 2015), employing identical parameters to process the inner information of each track separately.

**1) Global cross-track attention**. As shown in Fig. 3c(i), to achieve a globally consistent tempo across all instrument tracks, we introduce the global cross-track attention submodule. We slice the representation $z^s \in \mathbb{R}^{C \times T \times F}$ along the temporal dimension at index $t$ and along the frequency dimension at index $f$, obtaining $z_{t,f}^s \in \mathbb{R}^C$. Then we gather representations from all tracks across the temporal and frequency dimensions, denoted by $z_{1:T,1:F}^{1:S}$ and aggregate this information for $z_{t,f}^s$ as follows:

$$\text{Attn}_{\text{global\_cross}}(z_{t,f}^s) = \text{Attn}(W^{Q_1} z_{t,f}^s, W^{K_1} z_{1:T,1:F}^{1:S}, W^{V_1} z_{1:T,1:F}^{1:S}), \tag{4}$$

where $W^{Q_1}$, $W^{K_1}$, $W^{V_1} \in \mathbb{R}^C$ are learnable parameters in the global cross-track attention, and $\text{Attn}(\cdot)$ is the encoder layer of original Transformer. By allowing each track to reference the information of all tracks across both time and frequency dimensions, this approach helps maintain a consistent tempo across all instrument tracks.

**2) Time-specific cross-track attention**. While the global cross-track attention allows for consistent tempo across the entire piece, achieving finer-grained rhythmic synchronization between tracks requires localized temporal context for proper alignment. To this end, as shown in Fig. 3c(ii), we introduce a time-specific cross-track attention submodule. Taking the representation $z_{t,f}^s \in \mathbb{R}^C$ as an example. At the same $t$, we gather representation from all tracks across the frequency dimensions, denoted by $z_{t,1:F}^{1:S}$ and aggregate this information for $z_{t,f}^s \in \mathbb{R}^C$ as follows:

$$\text{Attn}_{\text{time\_cross}}(z_{t,f}^s) = \text{Attn}(W^{Q_2} z_{t,f}^s, W^{K_2} z_{t,1:F}^{1:S}, W^{V_2} z_{t,1:F}^{1:S}). \tag{5}$$

where $W^{Q_2}$, $W^{K_2}$ and $W^{V_2} \in \mathbb{R}^C$ are learnable parameters in the time-specific cross-track attention. By employing attention on other tracks at the same $t$, time-specific cross-track attention encourages instruments to align their musical events temporally, leading to tightly synchronized onset patterns and chordal structures.

**Track-specific modules**. In SyncTrack, we leverage the track-specific modules to capture track-specific information, such as divergent timbre and pitch range. As shown in Fig. 3b, we design a learnable instrument prior. First, we leverage one-hot vectors $V$ to represent different tracks. These vectors $V$ are encoded via positional encoding (Mildenhall et al., 2021) and subsequently transformed by a two-layer neural network. Finally, the embeddings of $V$ are added to the time embedding of $n$. Then, we add the learnable instrument prior to the output of the first ResBlock. The final track-specific representation is obtained from the second ResBlock.

## 4 METRICS FOR RHYTHMIC STABILITY AND SYNCHRONIZATION

To address the rhythmic stability and synchronization issues for multi-track music generation, we provide three different metrics. These metrics directly capture whether the beats are synchronized across all tracks $S$ for all samples $N$, offering a reproducible and interpretable way to assess multi-track music generation results across different methods.

**Inner-track Rhythmic Stability (IRS).** For music with a stable rhythm, beat intervals should remain consistent within each track. IRS quantifies temporal consistency by averaging the standard deviation of the Inter-Beat Interval (Dannenberg, 1987; Robertson, 2012) across all samples for each track $s$:

$$\text{IRS} = \mathbb{E}_{s,n} \left[ std(I_n^s) \right], \tag{6}$$

where $I_n^s$ denotes the beat intervals for the track $s$ in sample $n$, with each element defined as time difference between two consecutive beats.

**Cross-track Beat Synchronization (CBS)**. CBS measures rhythmic synchronization among multiple tracks. Inspired by the tolerance window concept in beat tracking (Dixon, 2001), we divide the timeline into multiple time windows and compute the proportion of tracks that contain at least one beat within each window. Only tracks that contain content are considered. CBS is defined as:

$$\text{CBS} = \mathbb{E}_n \left[ \frac{\sum_{i=1}^{T} r_{i,n}}{\sum_{i=1}^{T} \mathbb{I}(r_{i,n} > 0)} \right], \tag{7}$$

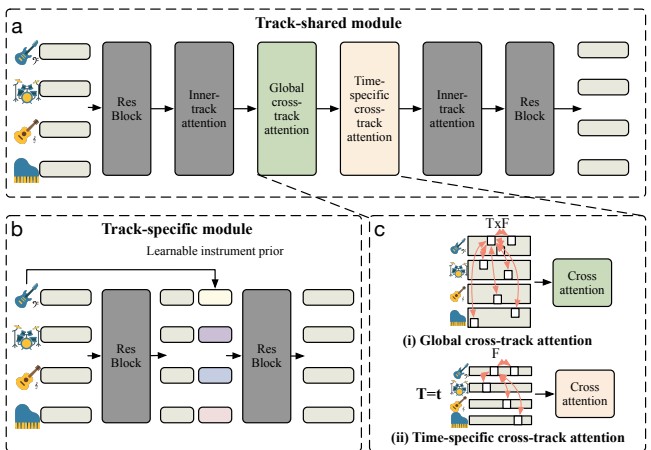

Figure 3: Illustration of the (a) track-shared module and (b) track-specific module. In (a), we leverage inner-track attention to capture the inner-track rhythmic stability and devise (c) two cross-track attention submodules to capture cross-track rhythmic stability and synchronization. In (b), we construct a learnable instrument prior to capture timbre and other track-specific features.

where $r_{i,n}$ is the ratio of tracks containing at least one beat within the $i$-th window, and we utilize $\mathbb{I}(r_{i,n} > 0)$ to exclude windows where no beat occurs in any track.

**Cross-track Beat Dispersion (CBD).** The CBD metric quantifies rhythmic synchronization in multi-track music by measuring the dispersion of beat alignment across all pairs of tracks. Inspired from GOTO's method (Goto & Muraoka, 1997), which evaluates alignment errors between estimated and reference beats using beat error sequences, CBD extends this concept to multi-track scenarios.

We select each track as reference in turn and compute the beat error sequence with respect to all other tracks. For track $s$ in sample $n$, let $b_{n,t}^s$ denote the $t$-th beat in the reference track. For each $b_{n,t}^s$, we find the matching beats in the other tracks and extract error sequence, denoted as $e(b_{n,t}^s)$. The CBD metric is defined as the mean or other statistics of the beat error sequence:

$$\text{CBD(mean)} = \mathbb{E}_{s,n,t}\left[e(b_{n,t}^s)\right]. \tag{8}$$

Note that, to eliminate the influence of tempo variations, we use the beat interval to normalize $e(b_{n,t}^s)$. Since the matching beats of $b_{n,t}^s$ are within the two intervals $\left[-I_{n,t-1}^s/2 + b_{n,t}^s, b_{n,t}^s\right]$ and $\left[b_{n,t}^s, b_{n,t}^s + I_{n,t}^s/2\right]$, we divide $e(b_{n,t}^s)$ by the corresponding interval length $I_{n,t-1}^s/2$ or $I_{n,t}^s/2$.

## 5 EXPERIMENTS

In this section, we conduct experiments to answer the following research questions:

*RQ1* Does SyncTrack outperform the state-of-the-art multi-track music generation methods?

*RQ2* Does SyncTrack perform well in inner-track stability and cross-track synchronization?

*RQ3* What are the respective contributions of the key modules to our method?

*RQ4* Are the proposed metrics robust, and can they reflect human subjective preference?

### 5.1 EXPERIMENTAL SETTING.

**Objective experiments.** We conduct objective experiments on Slakh2100 dataset (Manilow et al., 2019), following the common subset (Mariani et al., 2023) of four tracks: bass, drums, guitar, and piano. All audio files are resampled to 16 kHz and segmented into 10.24-second clips. Audio segments are converted into Mel-spectrograms for training. The model is initialized using pre-trained weights from MusicLDM (Chen et al., 2024) and trained for 320K iterations with a batch size of 16. Inference employs the DDIM sampler with 200 steps. For further details, see Appendix A.5.

**Subjective Experiments.** We conduct a web-based evaluation with two parts: mixture music assessment and individual track quality evaluation. For mixtures, we design four scoring groups, each having participants rate three audio samples comparing a ground-truth sample from Slakh2100 against generated samples from SyncTrack an MSG-LD on a 5-point scale. For individual tracks, we select two representative instruments: drums and piano, creating two scoring groups per instrument; each group presents three clips (from Slakh2100, SyncTrack, and MSG-LD) to be rated on a 3-point scale. Further details are in Appendix A.6.

**Baselines.** We choose MSDM (Mariani et al., 2023), STEMGEN (Parker et al., 2024), JEN-1 Composer (Yao et al., 2025b), and MSG-LD (Karchkhadze et al., 2025) as baselines. Our baseline selection is strictly limited to the task of multi-track waveform music generation. Models designed for stereo mixtures or symbolic music fall outside the scope of this comparison.

## 5.2 Quality of Generated Multi-track music (RQ1)

In multi-track music generation, the most widely used metric for evaluating overall audio quality is the FAD computed on the mixture of all tracks. We evaluate our SyncTrack in unconditional mode on the total music generation task using the Slakh test dataset. Then, we evaluate FAD between the generated mixtures and those from the test set. As shown in Table 1, SyncTrack significantly outperforms MSDM, resulting in a significant reduction in FAD scores from 6.55 to 1.26. Compared to other baselines, SyncTrack achieves reductions of 70.70%, 68.81% and 3.82% in FAD. This major improvement confirms that SyncTrack can generate high-quality, coherent mixture audio.

Table 1: FAD↓ scores of mixture music.

| Metrics | MSDM | STEMGEN | JEN-1 Composer | MSG-LD | SyncTrack |
|---|---|---|---|---|---|
| FAD | 6.55 | 4.3 | 4.04 | 1.31 | **1.26** |

We further evaluate FAD scores for each generated track, using MSDM and MSG-LD as baselines due to the availability of their open-source code. As shown in Table 2, SyncTrack consistently outperforms the baselines across all tracks, achieving FAD reductions of 32.38%, 27.55%, 20.77%, and 45.59% over MSG-LD for bass, drums, guitar, and piano, respectively.

Table 2: Track-wise FAD↓ scores.

| Method | Bass | Drum | Guitar | Piano |
|---|---|---|---|---|
| SyncTrack | **0.710** | **0.710** | **1.450** | **1.110** |
| MSG-LD | 1.050 | 0.980 | 1.830 | 2.040 |
| MSDM | 6.304 | 6.721 | 4.259 | 5.563 |

The lowest FAD scores are observed for rhythmic tracks like bass and drums, which exhibit the most distinctive percussive patterns, indicating our model's strength in capturing these grooves. For melodic tracks such as piano—with their broad pitch range and complex spectra—prior models have typically underperformed. The substantial FAD improvement (45.59%) on the Piano track highlights our model's enhanced ability to model the intricate distributions of challenging tracks.

Table 3: Subjective evaluation results.

| Method | Mixture | | | | Drum | | Guitar | |
|---|---|---|---|---|---|---|---|---|
| | 1 | 2 | 3 | 4 | 1 | 2 | 1 | 2 |
| Ground Truth | 4.2±0.9 | 4.5±0.6 | 4.7±0.5 | 4.6±0.6 | 3.0±0.2 | 2.6±0.7 | 2.9±0.3 | 3.0±0.2 |
| SyncTrack | 3.3±1.0 | 3.5±0.8 | 3.0±0.9 | 3.9±0.9 | 1.9±0.3 | 2.1±0.5 | 1.9±0.4 | 1.8±0.5 |
| MSG-LD | 1.5±0.6 | 1.3±0.5 | 1.8±0.9 | 1.7±0.8 | 1.2±0.5 | 1.3±0.6 | 1.2±0.5 | 1.2±0.4 |

The results of subjective experiments are shown in Table 3. The methods achieve the following average scores: Ground truth (4.48), MSG-LD (1.57), and SyncTrack (3.42), demonstrating that music generated by SyncTrack is better perceived by listeners.

## 5.3 RHYTHMIC STABILITY AND SYNCHRONIZATION (RQ2)

In this section, we investigate whether the quality improvement stems from enhanced rhythmic stability and synchronization. We evaluate the rhythmic stability (IRS) and cross-track rhythmic synchronization (CBD, CBS) of the music generated by SyncTrack and the baselines. Only statistical values are provided here. Further visualizations are provided in the Appendix A.3.

Table 4: Track-wise IRS↓ scores.

| Method | Bass | Drum | Guitar | Piano |
|---|---|---|---|---|
| Ground Truth | **0.015** | **0.005** | **0.016** | **0.015** |
| SyncTrack | **0.021** | **0.011** | **0.024** | **0.023** |
| MSG-LD | 0.041 | 0.040 | 0.039 | 0.039 |
| MSDM | 0.050 | 0.036 | 0.034 | 0.046 |

**Analysis of Inner Track Stability.** We evaluate rhythmic stability using our proposed IRS metric, which quantifies the variance of inter-beat intervals. A stable beat is essential for high-quality music, particularly for percussion instruments like drums. As shown in Table 4, ground truth tracks exhibit lower IRS values than all generated samples, with percussion achieving the lowest scores as expected. SyncTrack outperforms MSG-LD and MSDM in IRS across all tracks, showing the largest improvement on drums, which highlights its superior capability in modeling rhythmic patterns. Moreover, while FAD does not capture every aspect of quality, we can still observe that lower IRS is consistently associated with lower FAD, reflecting a reduced gap between generated and real music.

**Analysis of Cross-track Synchronization.** To analyze the rhythmic synchronization across tracks in multi-track music generation. We employ Cross-track Beat Synchronization (CBS) and Cross-track Beat Dispersion (CBD) as evaluation metrics. A higher CBS indicates more simultaneous beats, and a lower CBD reflects less dispersion, both implying stronger synchronization.

As shown in Table 5, SyncTrack achieves the best performance on both CBS and CBD metrics, indicating tighter rhythmic synchronization across tracks. For example, SyncTrack attains a CBS of 0.5206, which is 34.8% higher than MSG-LD (0.3861) and also outperforms MSDM (0.4694). Regarding CBD (mean), SyncTrack reaches 0.2681, representing a 27.8% reduction compared to MSG-LD (0.3714). SyncTrack also obtains the lowest CBD (std) and CBD (median), indicating the best rhythmic synchronization across tracks.

Table 5: Cross-track synchronization metrics scores.

| Metrics | Ground Truth | SyncTrack | MSG-LD | MSDM |
|---|---|---|---|---|
| CBS ↑ | 0.5740 | **0.5206** | 0.3861 | 0.4694 |
| CBD (mean)↓ | 0.2412 | **0.2681** | 0.3714 | 0.3127 |
| CBD (std)↓ | 0.1578 | **0.2131** | 0.2642 | 0.2217 |
| CBD (median)↓ | 0.2066 | **0.2258** | 0.3545 | 0.2811 |

## 5.4 ABLATION STUDY (RQ3)

SyncTrack is built upon the Backbone model by incorporating three key modules: ⓐ track-specific module, ⓑ global cross-track attention, ⓒ time-specific cross-track attention. Note that ⓑ and ⓒ appear sequentially within each track-shared module. We consider six ablation variants here: (1) Backbone; (2) Backbone w/ ⓐ; (3) Backbone w/ ⓐ+ⓑ; (4) Backbone w/ ⓐ+ⓒ; (5) SyncTrack-reorder, which reverses the order of these two attention modules ⓑ and ⓒ; (6) SyncTrack-alternate, which uses only one of ⓑ or ⓒ in each track-shared module alternatively. As shown in the Table 6, we report FAD scores for all models and the FAD improvement (Promotion) of SyncTrack's final mixed music over its ablated variants. Our experiments address the following questions:

1. Are all three modules useful? Yes. SyncTrack achieves significant improvement over the backbone (50%) and variants (11%-27%)

2. Do the three modules play distinct roles? The ablation studies confirm that each of the three modules serves a distinct and complementary function. Module ⓐ captures timbre and other track-specific features, resulting in a significant improvement in individual-track audio quality (ranging from 56.18% to 84.41%). Module ⓑ enhances the stability of each track. Hence, Backbone w/ ⓐ+ⓑ has significant improvement (6.3%-22.55%) over Backbone w/ ⓐ in single-track music quality. Module ⓒ enables fine-grained synchronization, aligning musical events across tracks at the same temporal position. Backbone w/ ⓐ+ⓒ shows significantly enhanced overall quality (17.97%) of multi-track music over the backbone w/ ⓐ.

3. Is the integration of modules well-reasoned? Yes. SyncTrack outperforms both SyncTrack-reorder and SyncTrack-alternate variants, supporting the design choice of incorporating both ⓑ and ⓒ concurrently as in the proposed order. Since ⓒ enables a more fine-grained synchronization building upon ⓑ, the sequence of first ⓑ followed by ⓒ is well-founded.

Table 6: Ablation study of track-specific module and two types of cross-track attention.

| Model | Bass | Drum | Guitar | Piano | Mixture | Promotion |
|---|---|---|---|---|---|---|
| Backbone | 5.234 | 3.081 | 6.012 | 6.170 | 2.570 | 50.97% |
| Backbone w/ ⓐ | 0.816 | 0.809 | 2.634 | 1.695 | 1.742 | 27.67% |
| Backbone w/ ⓐ+ⓑ | **0.632** | 0.758 | 2.367 | 1.359 | 1.627 | 22.56% |
| Backbone w/ ⓐ+ⓒ | 0.892 | 0.889 | 2.680 | 1.547 | 1.429 | 11.83% |
| SyncTrack-alternate | 0.900 | 0.897 | 2.663 | 1.757 | 1.586 | 20.55% |
| SyncTrack-reorder | 0.957 | 0.943 | 2.887 | 1.877 | 1.681 | 25.04% |
| SyncTrack | 0.710 | **0.710** | **1.450** | **1.110** | **1.260** | - |

## 5.5 ROBUSTNESS AND SUBJECTIVE VALIDATION OF EVALUATION METRICS(RQ4)

We select samples with varying objective metric scores (IRS, CBS, and CBD) and present them to human listeners for subjective evaluation. The results show a clear correspondence: samples with lower IRS receive higher subjective stability ratings, while those with higher CBS and lower CBD receive lower rhythmic synchronization scores, indicating that users perceive them as less rhythmically aligned. Experimental details and questionnaire settings are in Appendix A.6.

We further verify metric robustness by varying beat tracking hyperparameters; model rankings remain consistent across all settings, confirming the stability of our metrics (see Appendices A.1 and A.2).

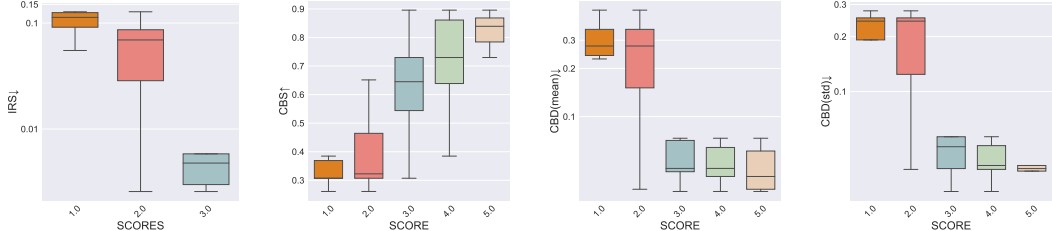

Figure 4: Comparison of subjective ratings and objective metric scores.

## 6 CONCLUSION

In conclusion, we present SyncTrack, which effectively addresses the often-overlooked issues of rhythmic stability and synchronization in multi-track music generation. By integrating track-shared modules that include cross-track attention submodules and track-specific modules that incorporate learnable instrument priors, SyncTrack is highly effective in achieving rhythmic consistency and capturing the unique characteristics of each track. Our proposed metrics—IRS, CBS, and CBD—offer a comprehensive framework for evaluating rhythmic features. Experiments show significant improvements in the musical and rhythmic quality of the music generated by our model. In the future, we plan to extend SyncTrack to generate longer-form multi-track music and broaden its applications.

## ACKNOWLEDGMENTS

This work was partially supported by an Area of Excellence project (AoE/E-601/24-N) and a Theme-based Research Project (T32-615/24-R) from the Research Grants Council of the Hong Kong Special Administrative Region, China. We also acknowledge the funding from the Hong Kong Innovation and Technology Commissionand (ITCPD/17-9). We would like to thank all reviewers for their helpful suggestions in improving this paper.

## ETHICS STATEMENT

This work presents a novel method in the field of machine learning. We have considered the potential societal impacts of our research. While our primary aim is to advance the state of knowledge, we acknowledge that any powerful technology could potentially be misused. We have no specific reason to believe that our work poses immediate significant risks, and we encourage the responsible use of the technology developed herein. All authors of this paper declare that there are no competing interests, financial or non-financial, related to this work.

## REPRODUCIBILITY STATEMENT

To ensure reproducibility, we provide comprehensive experimental details in Appendix A.5. Audio samples, alongside with the source code for both the model and evaluation metrics, are available on our demo page.

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

# A APPENDIX

## A.1 BEAT TRACKING IMPLEMENTATION DETAILS

**Tools and Default Settings.** We conduct objective experiments on Slakh2100 dataset (Manilow et al., 2019). Note that we do not choose MUSDB18 because data sparsity would lead to a large approximation error (Zhang et al., 2025). We utilize the *RNNDownBeatProcessor* and *DBNDownBeatTrackingProcessor* from the madmom (Böck et al., 2016) library for beat extraction throughout our experiments. Unless otherwise specified, we set the frame rate (fps) to 150Hz and used madmom's default transition lambda (tl) value (Böck et al., 2016) for all main results.

**Key Parameters.** Beat detection accuracy directly affects rhythm-related metrics. To examine the robustness of our metrics, we systematically varies two key parameters in beat tracking:

- *Frames per Second (fps)*: Higher fps provides finer temporal resolution, enabling more precise beat localization—especially important for multi-track or sparsely rhythmic content. Lower fps reduces computational cost but may degrade detection accuracy.
- *Transition Lambda (tl)*: This parameter controls temporal smoothing during beat sequence inference. Higher tl enforces smoother, more consistent tempo estimation, but may mask genuine rhythmic instabilities in generated tracks. Excessively high values can artificially inflate rhythmic stability scores by obscuring local irregularities. We find the default tl offered a good balance between sensitivity and smoothing, reliably reflecting true rhythmic quality.

## A.2 PARAMETER SENSITIVITY ANALYSIS

To rigorously assess the robustness of our evaluation metrics, we conduct experiments across a grid of beat tracking configurations, varying both fps and tl. Figures A1 and A2, as well as Table A1, summarizes the effects of these parameters on our three metrics: Inner-track Rhythmic Stability (IRS), Cross-track Beat Synchronization (CBS), and Cross-track Beat Dispersion (CBD).

Across all settings, the relative ranking of models remains stable, demonstrating that our metrics are robust to changes in beat tracking hyperparameters.

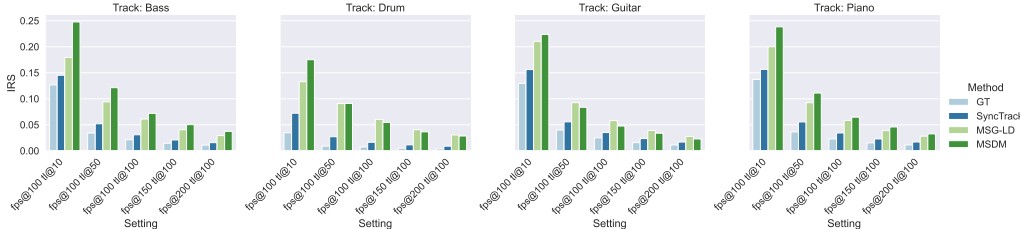

Figure A1: Comparison of IRS across hyperparameter settings,

**Inner-track Rhythmic Stability (IRS).** As shown in Figure A1, the relative ordering among GT, SyncTrack, MSG-LD, and MSDM remains unchanged as beat tracking parameters vary. This demonstrates that our rhythmic stability metric is robust to beat tracking hyperparameters, and that model comparisons are reliable regardless of the specific configuration.

Besides, increasing either fps or tl results in a consistent decrease in IRS values across all methods. This trend reflectes the influence of beat tracking hyperparameters: higher fps yield finer temporal resolution, allowing for more precise and stable beat localization, while higher tl enforces greater temporal smoothing and more regular tempo estimation. Both effects reduce the measured variance of beat intervals, thereby lowering IRS scores.

**Cross-track Beat Synchronization (CBS).** As shown in Figure A2, CBS exhibites a consistent dependency on both beat tracking hyperparameters and the chosen window size. Varying the window size has a clear and intuitive effect: smaller window sizes impose a stricter criterion for considering beats as synchronous, leading to lower CBS scores, whereas larger window sizes relax this criterion and yield higher CBS values. This relationship holds consistently for all methods and parameter settings.

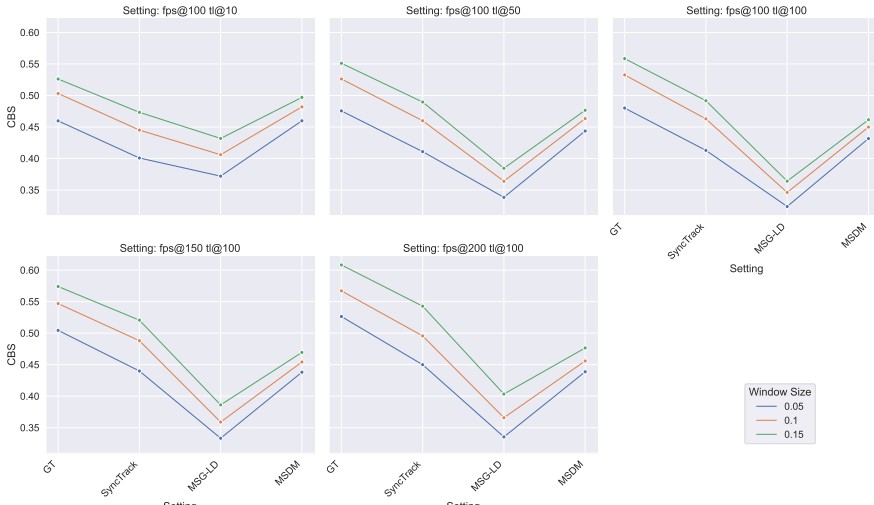

Figure A2: Comparison of CBS across hyperparameter settings.

Despite these variations in absolute CBS values, the relative ranking among GT, SyncTrack, MSG-LD, and MSDM remains unchanged across all configurations. Ground truth consistently achieves the highest CBS, and the ordering among models was preserved regardless of the hyperparameter or window size choices. This stability demonstrates the robustness of CBS as a comparative metric for evaluating model performance under diverse evaluation settings.

**Cross-track Beat Dispersion (CBD).** Table A1 presents detailed statistics for CBD mean, std, and median. Moreover, although the absolute values of CBD metrics varies slightly with fps and tl, the relative model ranking is almost unaffected. This further corroborates the insensitivity of our evaluation framework to beat-tracking hyperparameters.

Table A1: Comparison of CBD across hyperparameter settings.

| Setting | Metrics | Ground Truth | SyncTrack | MSG-LD | MSDM |
|---|---|---|---|---|---|
| fps@100 tl@50 | CBD (mean) ↓ | 0.2206 | **0.2589** | 0.3644 | 0.2923 |
| | CBD (std) ↓ | 0.1693 | **0.2241** | 0.2822 | 0.2370 |
| | CBD (median) ↓ | 0.1769 | **0.2058** | 0.3350 | 0.2424 |
| fps@100 tl@100 | CBD (mean) ↓ | 0.2143 | **0.2522** | 0.3829 | 0.3205 |
| | CBD (std) ↓ | 0.1556 | **0.2101** | 0.2708 | 0.2329 |
| | CBD (median) ↓ | 0.1762 | **0.2060** | 0.3679 | 0.2870 |
| fps@150 tl@100 | CBD (mean) ↓ | 0.2412 | **0.2681** | 0.3714 | 0.3127 |
| | CBD (std) ↓ | 0.1578 | **0.2131** | 0.2642 | 0.2217 |
| | CBD (median) ↓ | 0.2066 | **0.2258** | 0.3545 | 0.2811 |
| fps@200 tl@100 | CBD (mean) ↓ | 0.2642 | **0.2926** | 0.3590 | 0.3109 |
| | CBD (std) ↓ | 0.1639 | 0.2203 | 0.2534 | **0.2134** |
| | CBD (median) ↓ | 0.2316 | **0.2545** | 0.3407 | 0.2807 |

### A.3    IMPROVEMENT IN SYNCTRACK'S RHYTHMIC STABILITY AND SYNCHRONIZATION

We examine the consistency of this improvement in SyncTrack's Rhythmic Stability and Synchronization throughout the Slakh2100 dataset.

**Rhythmic Stability.** We examine the rhythmic stability by computing IRS scores for Bass, Drum, Guitar and Piano. From Figure A3, we can observe that SyncTrack achieves lowest IRS than MSDM and MSG-LD, demonstrating the generated music of SyncTrack closely approximates real music with greater rhythmic stability.

**Rhythmic Synchronization.** To analyzes the rhythmic synchronization across tracks in multi-track music generation. We employ CBS and CBD as evaluation metrics. CBS measures the proportion of time windows in which multiple tracks have simultaneous beats—a higher CBS indicates stronger synchronization. CBD quantifies the dispersion of beat timings across tracks—a lower CBD means greater rhythmic synchronization.

As shown in Table 4 in main text, SyncTrack achieves best performance in all generative methods. We further examine the consistency of this improvement in Rhythmic Synchronization. As shown in Figure A4, we can see the generated music of SyncTrack generally outperforms that of MSDM and MSG-LD. SyncTrack ensures superior rhythmic synchronization across multiple tracks.

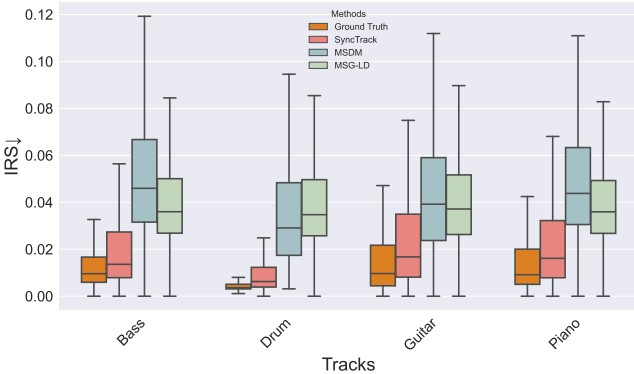

Figure A3: Track-wise IRS scores.

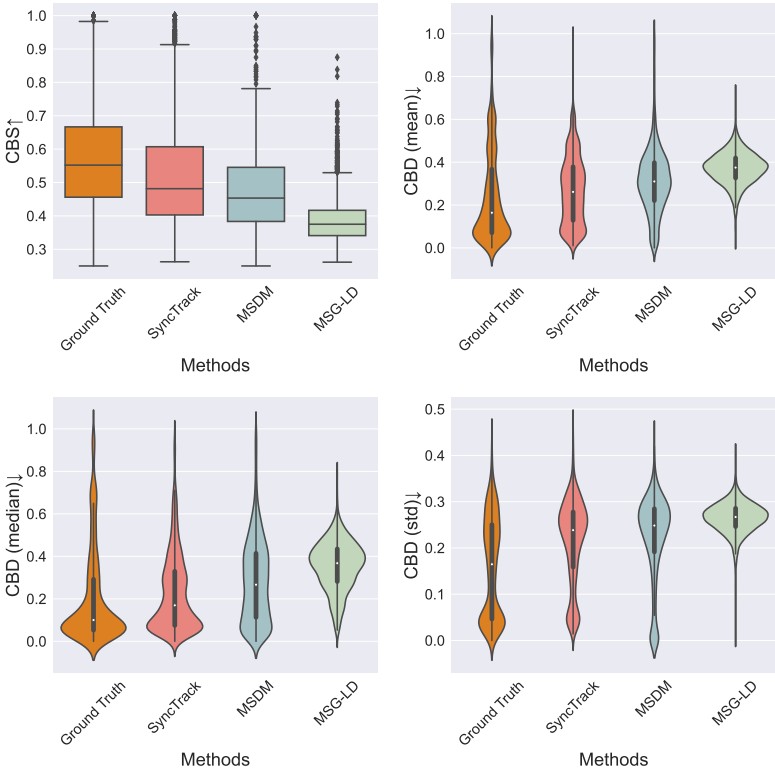

Figure A4: Cross-track synchronization metrics scores.

## A.4 CASE STUDY OF RHYTHM EVALUATION METRICS

To further validate the effectiveness and interpretability of our proposed rhythm evaluation metrics, we present a case study comparing both Ground Truth (GT) samples from the Slakh2100 dataset and generated samples from baseline models. We focus on three key metrics: IRS, CBS and CBD.

**Inner-track Rhythmic Stability.** Figure A5 showed two representative drum tracks from the GT dataset (left) and two from baseline-generated samples (right). For each, we visualize the spectrogram, beat and downbeat activations, and the extracted beat sequence. The GT drum tracks exhibit highly regular and stable beat intervals, as reflected in both the visualized beat grid and the very low IRS values (0.0049 and 0.0051). In contrast, the baseline-generated tracks display irregular beat sequences, with beat intervals that fluctuate over time and substantially higher IRS values (0.1859 and 0.1494). These results demonstrate that IRS effectively captures the stability of rhythmic patterns within a single track and aligned with intuitive human judgments.

**Multi-track Beat Synchronization.** We further analyze multi-track synchronization using two Ground Truth examples and two baseline-generated examples, as shown in Figure A6 and Table A2. The Ground Truth samples exhibit strong cross-track synchronization: beats from different instruments (bass, drums, guitar, and piano) are well aligned, as visible in the vertical alignment of beat annotations across tracks. Correspondingly, both CBS and CBD metrics indicate high synchronization and low dispersion across different track rhythm.

In contrast, the generated samples display chaotic beats across tracks, and in some cases, entire tracks are empty. This is reflected in the lower CBS values and higher CBD values, indicating both worse cross-track alignment and larger rhythmic dispersion between tracks.

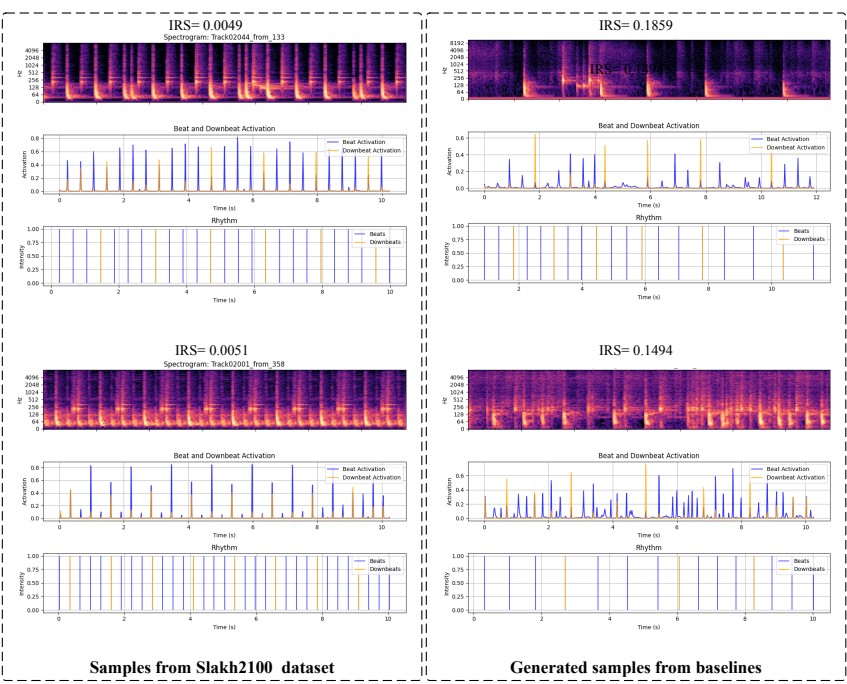

Figure A5: Case study of IRS for ground truth and generated samples.

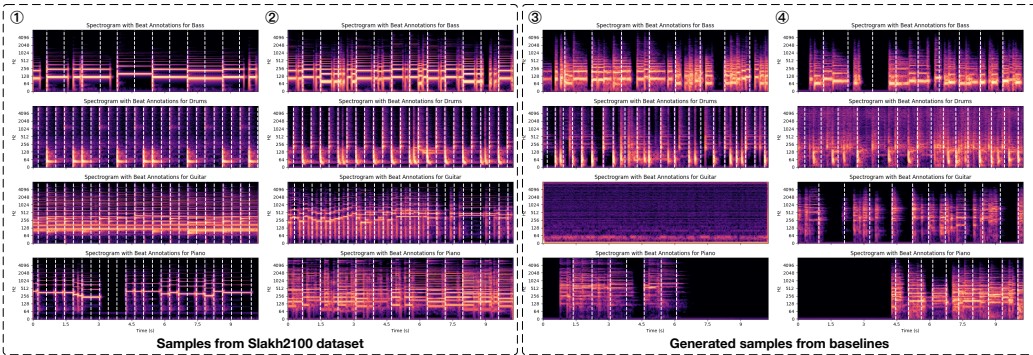

Figure A6: Visualization of cross-track synchronization in ground truth and generated samples.

Table A2: Comparison of CBS and CBD for ground truth and generated samples.

|  | Sample | CBD(mean) | CBD(std) | CBD(median) | CBS |
|---|---|---|---|---|---|
| Slakh2100 | ① | 0.0702 | 0.0481 | 0.0709 | 0.5286 |
|  | ② | 0.0628 | 0.0677 | 0.0553 | 0.7857 |
| Generated from baselines | ③ | 0.2619 | 0.2512 | 0.1522 | 0.3629 |
|  | ④ | 0.3975 | 0.2387 | 0.3815 | 0.3041 |

A.5    TRAINING CONFIGURATION AND MODEL DETAILS.

During training, the audio segments are converted into Mel-spectrograms using a window size of 1024 and a hop length of 160 samples. The mixture audio for each clip is obtained by summing the individual tracks. The Adam optimizer is used with a learning rate of $3e - 5$.

SyncTrack consists of 241M trainable parameters and 128M non-trainable parameters. More details of the architecture are summarized in Table A3. When trained on an A6000 GPU with a batch size of 16, each epoch takes approximately 11 minutes. The full training process required under 3.5 hours (3:07:37) to complete 21 epochs.

Table A3: Network architecture of SyncTrack

| Layer (type) | depth-idx | Input Shape | Output Shape | Param # |
|---|---|---|---|---|
| SyncTrack | - | [1, 4, 8, 64, 64] | - | 128,122,516 |
| - DiffusionWrapper | 1-1 | [1, 4, 8, 64, 64] | [1, 4, 8, 64, 64] | - |
| - UNetModel | 2-1 | [4, 16, 64, 64] | [4, 8, 64, 64] | 16 |
| - Sequential (time_embed) | 3-1 | [4, 128] | [4, 512] | - |
| - Linear | 4-1 | [4, 128] | [4, 512] | 66,048 |
| - SiLU | 4-2 | [4, 512] | [4, 512] | - |
| - Linear | 4-3 | [4, 512] | [4, 512] | 262,656 |
| - Sequential (label_emb) | 3-2 | [4, 8] | [4, 512] | - |
| - Linear | 4-4 | [4, 8] | [4, 128] | 1,152 |
| - SiLU | 4-5 | [4, 128] | [4, 128] | - |
| - Linear | 4-6 | [4, 128] | [4, 512] | 66,048 |
| - ModuleList (input_blocks) | 3-3 | - | - | - |
| - Track-specific Module | 4-7 | [4, 8, 64, 64] | [4, 128, 64, 64] | 9,344 |
| - Track-specific Module | 4-8 | [4, 128, 64, 64] | [4, 128, 64, 64] | 426,880 |
| - Track-specific Module | 4-9 | [4, 128, 64, 64] | [4, 128, 64, 64] | 426,880 |
| - Track-shared Module | 4-10 | [4, 128, 64, 64] | [4, 128, 32, 32] | 147,584 |
| - Track-shared Module | 4-11 | [4, 128, 32, 32] | [4, 256, 32, 32] | 3,681,280 |
| - Track-shared Module | 4-12 | [4, 256, 32, 32] | [4, 256, 32, 32] | 3,943,424 |
| - Track-shared Module | 4-13 | [4, 256, 32, 32] | [4, 256, 16, 16] | 590,080 |
| - Track-shared Module | 4-14 | [4, 256, 16, 16] | [4, 384, 16, 16] | 8,323,712 |
| - Track-shared Module | 4-15 | [4, 384, 16, 16] | [4, 384, 16, 16] | 8,667,648 |
| - Track-shared Module | 4-16 | [4, 384, 16, 16] | [4, 384, 8, 8] | 1,327,488 |
| - Track-shared Module | 4-17 | [4, 384, 8, 8] | [4, 640, 8, 8] | 22,392,448 |
| - Track-shared Module | 4-18 | [4, 640, 8, 8] | [4, 640, 8, 8] | 23,621,120 |
| - Track-shared Module (middle) | 3-4 | [4, 640, 8, 8] | [4, 640, 8, 8] | - |
| - Track-specific Module | 4-19 | [4, 640, 8, 8] | [4, 640, 8, 8] | 8,032,640 |
| - Track-shared Module | 4-20 | [4, 640, 8, 8] | [4, 640, 8, 8] | 15,588,480 |
| - Track-specific Module | 4-21 | [4, 640, 8, 8] | [4, 640, 8, 8] | 8,032,640 |
| - ModuleList (output_blocks) | 3-5 | - | - | - |
| - Track-shared Module | 4-22 | [4, 1280, 8, 8] | [4, 640, 8, 8] | 28,128,640 |
| - Track-shared Module | 4-23 | [4, 1280, 8, 8] | [4, 640, 8, 8] | 28,128,640 |
| - Track-shared Module | 4-24 | [4, 1024, 8, 8] | [4, 640, 16, 16] | 30,176,768 |
| - Track-shared Module | 4-25 | [4, 1024, 16, 16] | [4, 384, 16, 16] | 11,274,368 |
| - Track-shared Module | 4-26 | [4, 768, 16, 16] | [4, 384, 16, 16] | 10,290,816 |
| - Track-shared Module | 4-27 | [4, 640, 16, 16] | [4, 384, 32, 32] | 11,126,528 |
| - Track-shared Module | 4-28 | [4, 640, 32, 32] | [4, 256, 32, 32] | 4,993,024 |
| - Track-shared Module | 4-29 | [4, 512, 32, 32] | [4, 256, 32, 32] | 4,665,088 |
| - Track-shared Module | 4-30 | [4, 384, 32, 32] | [4, 256, 64, 64] | 4,927,232 |
| - Track-specific Module | 4-31 | [4, 384, 64, 64] | [4, 128, 64, 64] | 771,584 |
| - Track-specific Module | 4-32 | [4, 256, 64, 64] | [4, 128, 64, 64] | 607,488 |
| - Track-specific Module | 4-33 | [4, 256, 64, 64] | [4, 128, 64, 64] | 607,488 |
| - Sequential (conv_out) | 3-6 | [4, 128, 64, 64] | [4, 8, 64, 64] | - |
| - GroupNorm32 | 4-34 | [4, 128, 64, 64] | [4, 128, 64, 64] | 256 |
| - SiLU | 4-35 | [4, 128, 64, 64] | [4, 128, 64, 64] | - |
| - Conv2d | 4-36 | [4, 128, 64, 64] | [4, 8, 64, 64] | 9,224 |

### A.6 SUBJECTIVE EVALUATION.

To comprehensively validate the effectiveness of the proposed metrics, we aime to determine whether these objective indicators reflect human subjective perception. To achieve this, we conduct a web-based subjective evaluation consisting of two distinct experiments. We recruit 23 Participants to participate in the questionnaire survey.

**Experiment 1: Rhythmic Synchronization.** This experiment aims to evaluate the rhythmic synchronization across all tracks. We set up four scoring groups In each group, we place three audio excerpts sourced from Slakh2100, SyncTrack, and MSG-LD, with varying CBS/CBD values. These multi-track excerpts were mixed and presented to participants in a randomized order. Participants rated the degree of rhythmic synchronization among the instruments on a 5-point scale. The survey provided detailed explanations of the definition of synchronization and the meanings of each rating from 1 to 5.

**Experiment 2: Rhythmic Stability.** This experiment investigates the correlation between the proposed IRS metric and the human perceived rhythmic stability of individual track. We select two representative instruments: drums (rhythmic instrument) and piano (melodic instrument). Single-track excerpts—either the drum track or the piano track—with varying IRS values were sampled from ground truth data, generated music by our model and MSG-LD. Participants evaluate the inner-track rhythmic stability of each single-track excerpt using 3-point rating scale similar to Experiment 1.

### A.7 PROPOSED METRICS ON MUSDB18.

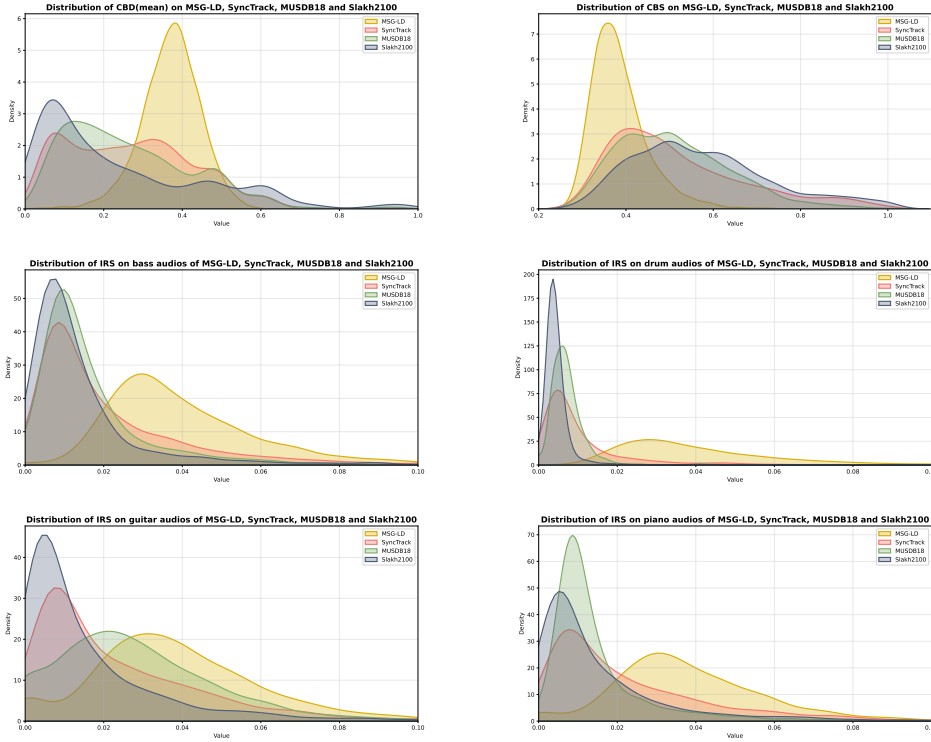

Figure A7: Distribution of three metrics on MSG-LD, SyncTrack, MUSDB18 and Slakh2100.

We utilize MUSDB18 (Rafii et al., 2017) to further illustrate the effectiveness of our proposed metrics. MUSDB18 comprises genuine studio recordings that encompass a wide range of musical styles. However, the inherent rhythmic variations and recording nuances in such real-world data can lead to less stable beat detection, which in turn results in its performance on rhythm synchronization metrics being lower than that of the perfectly aligned Slakh2100 dataset. Table A4 also demonstrates that the CBS, CBD, and IRS scores on MUSDB18 are slightly lower than those on Slakh2100. However,

real world music still outperforms generated music, and our results are fully consistent with this intuition. Furthermore, we demonstrate the distributions of the three proposed metrics on music from Slakh2100, MUSDB18 and the model generated music (MSDM, MSG-LD and SyncTrack) in Fig. A7. It can be observed that the distributions of the three proposed metrics on music from Slakh2100 are better than those on MUSDB18. Despite this, MUSDB18 maintains overall better harmony and rhythmic stability than music generated by both MSG-LD and SyncTrack.

Table A4: Scores on three metrics for the MUSDB18 and Slakh2100

| | CBS↑ | CBD(mean)↓ | CBD(std)↓ | CBD(median)↓ | IRS(B)↓ | IRS(D)↓ | IRS(G)↓ | IRS(P)↓ |
|---|---|---|---|---|---|---|---|---|
| MSG-LD | 0.386 | 0.371 | 0.264 | 0.355 | 0.041 | 0.040 | 0.039 | 0.039 |
| MSDM | 0.469 | 0.313 | 0.222 | 0.281 | 0.050 | 0.036 | 0.034 | 0.046 |
| SyncTrack | 0.521 | 0.268 | 0.213 | 0.226 | 0.021 | 0.011 | 0.024 | 0.023 |
| MUSDB18 | 0.512 | 0.264 | 0.197 | 0.223 | 0.017 | 0.007 | 0.029 | 0.015 |
| Slakh2100 | 0.574 | 0.241 | 0.158 | 0.207 | 0.015 | 0.005 | 0.016 | 0.015 |

## A.8 PROPOSED METRICS ON CONDITIONAL GENERATION

CBS/CBD still can apply in the conditional generation. Conditional generation refers to the task of generating the remaining tracks given a subset of known tracks, thereby completing a coherent multi-track arrangement. For the four-track (bass, drums, guitar, and piano) music generation task, we test four scenarios: (1) B: generation of Bass given the other three tracks; (2) BP: generation of Bass and Piano given the other two tracks; (3) BGP: generation of Bass, Guitar, and Piano; (4) BDGP: generation of all four tracks together.

As shown in the Table A5, the CBS and CBD metrics can still be computed for all these cases. Furthermore, the metric results align with intuition. Based on task difficulty, rhythm synchronization should follow BGDP>BGP>BP>B. According to our tests, the obtained CBS and CBD scores conform to this ordering.

Table A5: CBS and CBD metrics across different generation scenarios.

| | GT | B | BP | BGP | BDGP |
|---|---|---|---|---|---|
| CBS↑ | 0.5740 | 0.5620 | 0.5576 | 0.5259 | 0.5206 |
| CBD(std)↓ | 0.1578 | 0.1902 | 0.1940 | 0.2118 | 0.2131 |
| CBD(median)↓ | 0.2066 | 0.2192 | 0.2188 | 0.2304 | 0.2258 |
| CBD(mean)↓ | 0.2412 | 0.2591 | 0.2579 | 0.2718 | 0.2681 |

## A.9 PROPOSED METRICS ON DIFFERENT SEGMENT LENGTH

We examine the effectiveness of our metrics on segments of lengths 10s, 30s, 60s, and full songs (261.83±59.96s). As shown in Table A6, we can see that our our metrics are in all cases. Moreover, as the segment length increases, rhythm synchronization becomes more prone to degradation. According to our tests, the obtained CBS and CBD scores conform to this ordering.

Furthermore, we present the distributions of three metrics for music pieces of different lengths in Figure A8. Intuitively, for short music pieces like 10s, metrics on most samples are well, but there are still some poorer samples. For long music pieces, i.e. full song, the metrics are relatively stable across all samples.

Table A6: Slakh2100 scores on different audio segment length

| Segment length | CBS↑ | CBD↓ | | | IRS↓ | | | |
|---|---|---|---|---|---|---|---|---|
| | | mean | std | median | Bass | Drum | Guitar | Piano |
| 10s | 0.5740 | 0.2412 | 0.1578 | 0.2066 | 0.015 | 0.005 | 0.016 | 0.015 |
| 30s | 0.5554 | 0.2344 | 0.1742 | 0.1922 | 0.0203 | 0.007 | 0.0298 | 0.0254 |
| 60s | 0.5349 | 0.2471 | 0.1960 | 0.2018 | 0.0258 | 0.0121 | 0.0425 | 0.0366 |
| full-song | 0.4844 | 0.2660 | 0.2466 | 0.1995 | 0.0489 | 0.0273 | 0.0671 | 0.0710 |

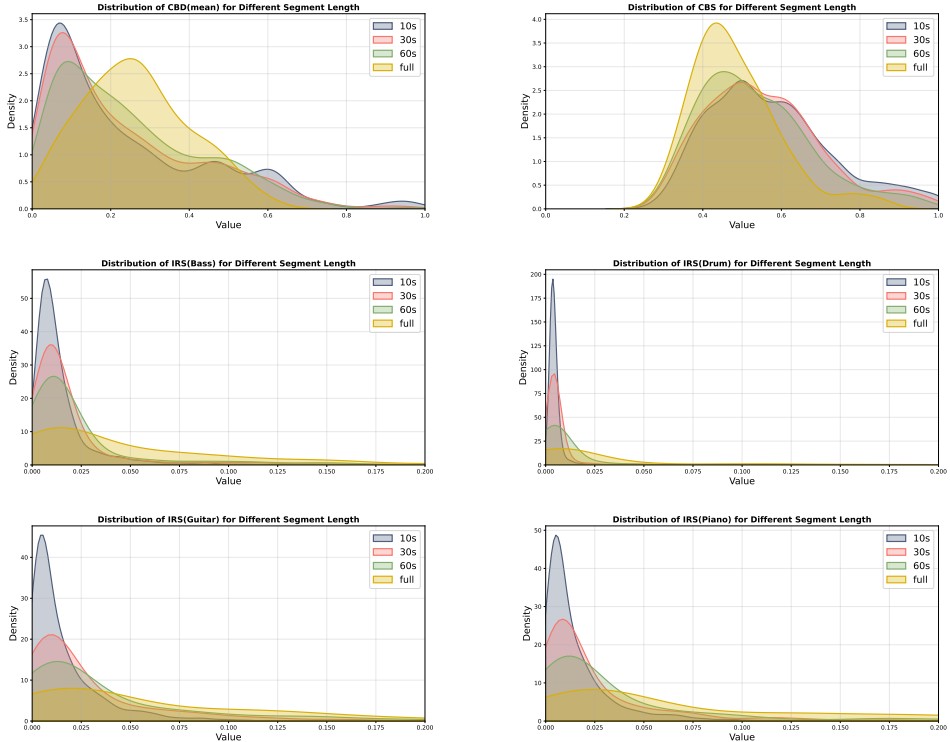

Figure A8: Distribution of three metrics on different segment length.

## A.10 THE USE OF LARGE LANGUAGE MODELS(LLMS)

In the preparation of this work, the authors used GPT-4 (Achiam et al., 2023) to polish and improve the clarity of the English text. All generated content was carefully reviewed, edited, and critically evaluated by the authors. The core ideas, experimental design, data analysis, result interpretation, and the final content of the paper remain the sole responsibility of the authors. The LLM served solely as a writing assistance tool and did not replace the authors' critical thinking or academic judgment.

