# OpenReview forum: "SyncTrack: Rhythmic Stability and Synchronization in Multi-Track Music Generation"
_ICLR.cc/2026/Conference — ICLR 2026 Poster_

### Official Review · Reviewer_1JZQ · 2025-10-29

**Soundness:** 3
**Presentation:** 3
**Contribution:** 2
**Rating:** 4
**Confidence:** 3

**Summary:**

In this paper, the authors propose a model architecture where modules are connected to allow global and local cross-track attention. This resulted in improved scores in terms of FAD as well as the proposed CSB metric.

**Strengths:**

The core idea of the paper of using cross-attention globally & time-specifically is simple and intuitive. Based on a holistic evaluation by FAD and the specific evaluation by IRS, the proposed idea seems to achieve the goal. The paper also presents additional experiment results with ablation and subjective tests.

**Weaknesses:**

The background sections can be improved. It seems to cover the music generation well, but synchronized generation is an active research topic in audio generation in the context of videos.

The proposed process can be explained in more detail. The paper covers the high-level information well, but it lacks of the detailed operations.

It is nice to see a new proposal on evaluation metrics, but it needs more research and experiment to justify them.

**Questions:**

It would be much nicer if the proposed method is discussed in more detail. For example, from Eqs. (4-5), what is the dimensions of the W variables? What is the exact operation of Eq(4), especially Attn()? These two equations are too important to be discussed on this high level only. This also raise an issue of reproducibility: I'm not sure if the paper has enough details for the readers to reproduce the model.

I'm also curious and a bit concerned about the proposed metrics - CBS (mean, std, median) and CBD. Are they really ↑ to 1 and ↓ to 0? It may feel deviated from the original idea of the paper, but since these are proposed by the authors in this paper, I wonder what is the reasonable range of them, perhaps measured on real human composed and produced music signals.

On FAD, SyncTrack indeed shows the lowest FAD scores, but why? The proposed method is about time sync. Why does it lead to a lower FAD?

Also on FAD, I would like more details. What was the model or source code used to measure it?

There are minor typeos, e.g. L159 - "estimator,offering". L441 - "Backbone w track".

---

> ### Author Response · Authors · 2025-11-21
> **Response to Reviewer 1JZQ (1/3)**
>
> We sincerely thank you for your encouraging and perceptive feedback, which has helped us address key aspects of our methodology and evaluation. We have carefully addressed each of your comments and concerns, as detailed below.
>
> ---
>
> **W1: The background sections can be improved. It seems to cover the music generation well, but synchronized generation is an active research topic in audio generation in the context of videos.**
>
> > **Response:** Thank you for this valuable suggestion! We have expanded the background section to cover synchronized generation in video-to-audio contexts. The additions encompass:
> >
> > 1. **An overview of methods:** We summarize recent approaches that achieve synchronization,including Diff-BGM, LORIS, VidMusician, VidMuse, and Mel-QCD [1-5], etc.
> > 2. **Discussion on evaluation metrics:** We introduce the common objective evaluation metrics for assessing rhythm synchronization used in this field (BCS, BHS) [2,3], analyzing their limitations for multi-track music generation due to coarse granularity.
> >
> > We hope these additions can better position our work within the broader context of synchronized generation research and clarify the distinct challenges in multi-track music synchronization task.
>
> >[1] Sizhe Li, Yiming Qin, Minghang Zheng, Xin Jin, and Yang Liu. Diff-bgm: A diffusion model for video background music generation. In Proceedings of the IEEE/CVF Conference on Computer Vision and Pattern Recognition, pp. 27348–27357, 2024b.
> >
> >[2] Jiashuo Yu, Yaohui Wang, Xinyuan Chen, Xiao Sun, and Yu Qiao. Long-term rhythmic video soundtracker. In International Conference on Machine Learning, pp. 40339–40353. PMLR, 2023.
> >
> >[3] Sifei Li, Binxin Yang, Chunji Yin, Chong Sun, Yuxin Zhang, Weiming Dong, and Chen Li. Vidmusician: Video-to-music generation with semantic-rhythmic alignment via hierarchical visual features. arXiv preprint arXiv:2412.06296, 2024a.
> >
> >[4] Zeyue Tian, Zhaoyang Liu, Ruibin Yuan, Jiahao Pan, Qifeng Liu, Xu Tan, Qifeng Chen, Wei Xue, and Yike Guo. Vidmuse: A simple video-to-music generation framework with long-short-term modeling. In Proceedings of the Computer Vision and Pattern Recognition Conference, pp. 18782–18793, 2025.
> >
> >[5] Juncheng Wang, Chao Xu, Cheng Yu, Lei Shang, Zhe Hu, Shujun Wang, and Liefeng Bo. Synchronized video-to-audio generation via mel quantization-continuum decomposition. In Proceedings of the Computer Vision and Pattern Recognition Conference, pp. 3111–3120, 2025.
>
>
> > **Modification in the manuscript:**  Section 2.3 in the main text.
>
>
> **W2 \& Q1: The proposed process can be explained in more detail. The paper covers the high-level information well, but it lacks of the detailed operations. It would be much nicer if the proposed method is discussed in more detail. This also raise an issue of reproducibility: I'm not sure if the paper has enough details for the readers to reproduce the model.**
>
> >**Response:** We appreciate your feedback regarding the level of detail and concerns of reproducibility. To address this, we have added comprehensive explanations of the dimensions of the W variables and the exact operation of the Attn($\cdot$) function in the main text. And we also have included the detailed model specifications in the supplementary materials (Table A3). Additionally, to ensure full reproducibility, we have provided a link to our anonymous code in the abstract.
>
>
> > **Modification in the manuscript:** Section 3 in the main text, Section A.5, Table A3 in the supplementary.

---

> ### Author Response · Authors · 2025-11-21
> **Response to Reviewer 1JZQ (2/3)**
>
> **W3: It is nice to see a new proposal on evaluation metrics, but it needs more research and experiment to justify them.**
>
> >**Response:** We sincerely thank the reviewer for this insightful feedback. To thoroughly validate the robustness and general applicability of our proposed metrics, we have conducted extensive additional experiments focusing on three critical dimensions: genre diversity, segment length, and conditional generation scenarios. The results consistently confirm that our metrics are both reliable and widely applicable.
> >
> >>**1. Robustness Across Music Genres**
> >>
> >> We aim to evaluate our metrics on more challenging and diverse music dataset. MUSDB18 comprises genuine studio recordings that encompass a wide range of musical styles. However, the inherent rhythmic variations and recording nuances in such real-world data can lead to less stable beat detection, which should result in its performance on rhythm synchronization metrics being lower than that of the perfectly aligned Slakh2100 dataset.
> >> As anticipated, the table below demonstrates that the CBS, CBD, and IRS scores on MUSDB18 are slightly lower than those on Slakh2100. Nevertheless, the real music in MUSDB18 still surpasses the generated music in our metrics, affirming their ability to reflect meaningful qualitative differences.
>
>
>
>
> | Model/Dataset | CBS↑ | CBD(mean)↓ | CBD(std)↓ | CBD(median)↓ | IRS(B)↓ | IRS(D)↓ | IRS(G)↓ | IRS(P\)↓ |
> |---------------|------|------------|-----------|--------------|---------|---------|---------|---------|
> | MSG-LD        | 0.386 | 0.371      | 0.264     | 0.355        | 0.041   | 0.040   | 0.039   | 0.039   |
> | MSDM          | 0.469 | 0.313      | 0.222     | 0.281        | 0.050   | 0.036   | 0.034   | 0.046   |
> | SyncTrack     | 0.521 | 0.268      | 0.213     | 0.226        | 0.021   | 0.011   | 0.024   | 0.023   |
> | MUSDB18       | 0.512 | 0.264      | 0.197     | 0.223        | 0.017   | 0.007   | 0.029   | 0.015   |
> | Slakh2100     | 0.574 | 0.241      | 0.158     | 0.207        | 0.015   | 0.005   | 0.016   | 0.015   |
>
> *Table 1: Scores on three metrics for the MUSDB18 and Slakh2100.*
>
> > **Modification in the manuscript:** Section A.7 in the supplementary.
>
>
> >> **2. Independence from Segment Length**
> >> We further assess metric stability across varying audio segment lengths: 10s, 30s, 60s, and full songs (261.83±59.96s). Intuitively, the longer the music segment, the more likely it is to contain sparse or chordal content, which in turn degrades both rhythm stability and synchronization. As shown in Table 2, all metrics remain computable and meaningful at every segment length. And longer segments exhibit slightly degraded scores. Hence, our metrics are not only length-agnostic but also perceptually grounded.
>
>
> | Segments length   | CBS↑    | CBD(mean)↓ | CBD(std)↓ | CBD(median)↓ | IRS(B)↓ | IRS(D)↓ | IRS(G)↓ | IRS(P\)↓ |
> |------------|---------|------------|-----------|--------------|---------|---------|---------|---------|
> | 10s        | 0.5740  | 0.2412     | 0.1578    | 0.2066       | 0.015   | 0.005   | 0.016   | 0.015   |
> | 30s        | 0.5554  | 0.2344     | 0.1742    | 0.1922       | 0.0203  | 0.007   | 0.0298  | 0.0254  |
> | 60s        | 0.5349  | 0.2471     | 0.1960    | 0.2018       | 0.0258  | 0.0121  | 0.0425  | 0.0366  |
> | full-song | 0.4844  | 0.2660     | 0.2466    | 0.1995       | 0.0489  | 0.0273  | 0.0671  | 0.0710  |
>
> *Table 2: Slakh2100 scores on different audio segment length*
>
> >**Modification in the manuscript**: Section A.9 in the supplementary.
>
> >> **3. Applicability to Conditional Generation**
> >>
> >>Beyond full-track generation, we validate our metrics in conditional generation settings, where only a subset of tracks is generated given the others. We test four scenarios with progressively increasing generation difficulty: (1) B: generation of Bass given the other three tracks; (2) BP: generation of Bass and Piano given the other two tracks; (3) BGP: generation of Bass, Guitar, and Piano given Drum track; (4) BDGP: generation of all four tracks together. The CBS and CBD scores in Table 3 closely match the expected difficulty ordering (**BGDP $>$ BGP $>$ BP $>$ B**), thereby confirming the metrics' validity for conditional generation tasks.
>
>
> |              | GT     | B   | BP  | BGP  | BDGP   |
> | ------------ | ------ | ------ | ------ | ------ | ------ |
> | CBS↑        | 0.5740 | 0.5620 | 0.5576 | 0.5259 | 0.5206 |
> | CBD(std)↓    | 0.1578 | 0.1902 | 0.1940 | 0.2118 | 0.2131 |
> | CBD(median)↓ | 0.2066 | 0.2192 | 0.2188 | 0.2304 | 0.2258 |
> | CBD(mean)↓    | 0.2412 | 0.2591 | 0.2579 | 0.2718 | 0.2681 |
>
> *Table 3: CBS and CBD metrics across different generation scenarios.*
>
> > **Modification in the manuscript**: Section A.8 in the supplementary.

---

> ### Author Response · Authors · 2025-11-21
> **Response to Reviewer 1JZQ (3/3)**
>
> **Q2: I'm also curious and a bit concerned about the proposed metrics - CBS and CBD (mean, std, median). Are they really up to 1 and down to 0? It may feel deviated from the original idea of the paper, but since these are proposed by the authors in this paper, I wonder what is the reasonable range of them, perhaps measured on real human composed and produced music signals.**
>
> > **Response:** Thank you for this question. CBS measures the proportion of tracks that contain at least one beat within each window. Hence, the range of CBS is from 1/N to 1, where N is the number of tracks. CBD quantifies the dispersion of beat alignment across all track pairs, with a minimum value of 0 for perfectly aligned music.
> >
> > Based on our evaluations across different real human-composed music datasets (**MUSDB18** and **Slakh2100**), and under our hyperparameter settings, music with **CBD(mean) < 0.27**, **CBD(std) < 0.2**, and **CBD(median) < 0.23** exhibits relatively good rhythm synchronization. We also present the plots of distribution of our three proposed metrics across MSG-LD, SyncTrack, MUSDB18, and Slakh2100. Detailed results are provided in the supplementary materials.
>
> > **Modification in the manuscript**: Section A.7 in the supplementary.
>
> **Q3: On FAD, SyncTrack indeed shows the lowest FAD scores, but why? The proposed method is about time sync. Why does it lead to a lower FAD? Also on FAD, I would like more details. What was the model or source code used to measure it?**
> > **Response:** Thank you for raising these points. Following the methodologies of MSDM [1] and MSG-LD [2], we trained SyncTrack on the Slakh2100 training set. For evaluation, we computed the FAD between the generated music and the Slakh2100 test set. We used the default backbone from the `audioldm_eval`$^1$  Python package to encode all audio samples into embeddings, which follows calculation protocol from MSDM [1]. The feature distributions of the generated music and the test set were modeled as multivariate Gaussians, and the FAD was computed as the Fréchet Distance between these distributions.
> >
> > Fundamentally, FAD measures the similarity between the generated music and the ground truth. The test set of Slakh2100 is utilized as the ground truth. Slakh2100 exhibits extremely strong rhythmic consistency. Our method produces results with superior time synchronization, making them closer to the real Slakh2100 data distribution, which consequently leads to a lower FAD score.
>
> >[1] Mariani G, Tallini I, Postolache E, et al. Multi-Source Diffusion Models for Simultaneous Music Generation and Separation[C]//ICLR. 2024.
> >
> >[2] Karchkhadze T, Izadi M R, Dubnov S. Simultaneous music separation and generation using multi-track latent diffusion models[C]//ICASSP 2025-2025 IEEE International Conference on Acoustics, Speech and Signal Processing (ICASSP). IEEE, 2025: 1-5.
> >1.https://github.com/haoheliu/audioldm_eval
>
> **Q4: There are minor typos, e.g. L159 - "estimator,offering". L441 - "Backbone w track"**
>
> >Thank you for pointing out these errors. We have corrected the typos and have thoroughly proofread the entire manuscript to ensure no similar mistakes remian.
> ---
> Your thoughtful observations have been essential in refining our work, and we are grateful for your support and insightful recommendations! We hope our responses and revisions can adequately address your concerns.

---

### Official Review · Reviewer_CJkQ · 2025-10-30

**Soundness:** 3
**Presentation:** 3
**Contribution:** 2
**Rating:** 6
**Confidence:** 3

**Summary:**

This paper presents SyncTrack, a latent diffusion framework for unconditional multi-track music generation designed to enhance rhythmic stability and cross-track synchronization. The model integrates global cross-track attention with time-aligned cross-track attention, along with a track-specific (instrument-prior) branch that maintains timbral distinctions across instruments. The authors further introduce three reproducible evaluation metrics—IRS (intra-track rhythmic stability) and CBS/CBD (cross-track synchronization/dispersion)—and conduct experiments on Slakh2100 using a four-track configuration (16 kHz, ~10.24 s segments, ~200 DDIM steps). The proposed method achieves improved FAD (both mix-level and per-track) compared with waveform-based baselines, and the CBS↑/CBD↓ trends align with human listening evaluations.

**Strengths:**

1. The paper introduces clear and reproducible rhythm-centric evaluation metrics (IRS/CBS/CBD) and provides robustness analyses, making these metrics potentially useful for broader community adoption.
2. The results demonstrate good alignment between objective and subjective evaluations: FAD improvements are consistent with listening preferences, and component-level ablations further validate the architectural design.

**Weaknesses:**

1. The current setup (10.24 s at 16 kHz) constrains both long-range musical structure and high-frequency detail. Including evaluations on longer segments (≥30–60 s) or full-song contexts would strengthen the claims.
2. Key details such as the number of participants, votes per sample, confidence intervals, or statistical significance, and loudness normalization procedures should be clearly reported in the main paper.

**Questions:**

1. Could the authors clarify the listening-test protocol in the main text, including details such as the panel size, number of votes per item, statistical testing procedures, and loudness normalization, to improve transparency and reproducibility?

2. Could the authors provide a detailed efficiency and time analysis?

---

> ### Author Response · Authors · 2025-11-21
> **Response to Reviewer CJkQ (1/2)**
>
> We sincerely thank you for your thoughtful and constructive feedback, which has helped us improve the clarity and completeness of our work. Below, we provide point-by-point responses to your comments and questions.
>
> ---
> **W1: The current setup (10.24 s at 16 kHz) constrains both long-range musical structure and high-frequency detail. Including evaluations on longer segments (30\-60 s) or full-song contexts would strengthen the claims.**
>
> >**Response:** Following your constructive comment. We have evaluated the effectiveness of our proposed metrics on the Slakh2100 test data sliced into segments of lengths **10s**, **30s**, **60s**, and **full songs (261.83±59.96s)**. The results demonstrate that our metrics remain applicable across all segment lengths. Moreover, the longer the music segment, the more likely it is to contain sparse or chordal content, which in turn degrades both rhythm stability and synchronization. Our results also show that the CBS, CBD, and IRS scores for full songs are slightly lower than those obtained on 10s segments.
> >
> > We add these experiments and present the distributions of three metrics for music pieces across different lengths in the supplementary material (Section A.9). Thank you again for your valuable input, which has helped us broaden the applicability of CBS and CBD.
>
>
> | Segments length   | CBS↑    | CBD(mean)↓ | CBD(std)↓ | CBD(median)↓ | IRS(B)↓ | IRS(D)↓ | IRS(G)↓ | IRS(P\)↓ |
> |------------|---------|------------|-----------|--------------|---------|---------|---------|---------|
> | 10s        | 0.5740  | 0.2412     | 0.1578    | 0.2066       | 0.015   | 0.005   | 0.016   | 0.015   |
> | 30s        | 0.5554  | 0.2344     | 0.1742    | 0.1922       | 0.0203  | 0.007   | 0.0298  | 0.0254  |
> | 60s        | 0.5349  | 0.2471     | 0.1960    | 0.2018       | 0.0258  | 0.0121  | 0.0425  | 0.0366  |
> | full-song | 0.4844  | 0.2660     | 0.2466    | 0.1995       | 0.0489  | 0.0273  | 0.0671  | 0.0710  |
>
> *Table 1: Slakh2100 scores on different audio segment lengths*
>
> >**Modification in the manuscript**: Section A.9 in the supplementary.

---

> ### Author Response · Authors · 2025-11-21
> **Response to Reviewer CJkQ (2/2)**
>
> **W2 \& Q1: Key details such as the number of participants, votes per sample, confidence intervals, or statistical significance, and loudness normalization procedures should be clearly reported in the main paper. Could the authors clarify the listening-test protocol in the main text, including details such as the panel size, number of votes per item, statistical testing procedures, and loudness normalization, to improve transparency and reproducibility?**
> >
> >**Response:** Thank you for highlighting the need for greater transparency. To address this, we have expanded the description of the subjective evaluation with further methodological details and relocated it from the supplementary to the main text (**Section 5.1**).
> >
> >We conduct a web-based evaluation with two parts: mixture music assessment and individual track quality evaluation. For mixtures, we design four scoring groups, each having participants rate three audio samples comparing a ground-truth sample from Slakh2100 against generated samples from SyncTrack an MSG-LD on a 5-point scale. For individual tracks, we select two representative instruments: drums and piano, creating two scoring groups per instrument; each group presents three clips (from Slakh2100, SyncTrack, and MSG-LD) to be rated on a 3-point scale.
> >
> >The table below summarizes the subjective evaluation results (mean ± standard deviation) of the scores in each group, now included as **Table 3** in the main text.
> >
> > Additionally, all audio samples are loudness-normalized prior to evaluation to ensure a consistent listening experience. To further support reproducibility, we have included the complete per-participant scoring data in the supplementary materials.
>
>
> | Method      | Mixture 1     | Mixture 2     | Mixture 3     | Mixture 4     | Drum 1       | Drum 2       | Guitar 1     | Guitar 2     |
> |-------------|---------------|---------------|---------------|---------------|--------------|--------------|--------------|--------------|
> | Ground Truth | 4.2±0.9      | 4.5±0.6      | 4.7±0.5      | 4.6±0.6      | 3.0±0.2     | 2.6±0.7     | 2.9±0.3     | 3.0±0.2     |
> | SyncTrack   | 3.3±1.0      | 3.5±0.8      | 3.0±0.9      | 3.9±0.9      | 1.9±0.3     | 2.1±0.5     | 1.9±0.4     | 1.8±0.5     |
> | MSG-LD      | 1.5±0.6      | 1.3±0.5      | 1.8±0.9      | 1.7±0.8      | 1.2±0.5     | 1.3±0.6     | 1.2±0.5     | 1.2±0.4     |
>
> *Table 2：Subjective evaluation results.*
>
> >**Modification in the manuscript**: Section 5.1 and Table 3 in the main text
>
>
> **Q2: Could the authors provide a detailed efficiency and time analysis?**
>
> > **Response:** Thank you for your considerate suggestion. We have now included a detailed efficiency analysis in the supplementary material (**Section A.5**). SyncTrack comprises 241M trainable parameters and 128M non-trainable parameters. When trained on an A6000 GPU with a batch size of 16, each epoch takes approximately 11 minutes. The full training process required under 3.5 hours (3:07:37) to complete 21 epochs.
>
> > **Modification in the manuscript:** Section A.5 in the supplementary.
>
> ---
>
> Thank you again for your careful review and for helping us present our research more comprehensively.

---

> > ### Comment · Reviewer_CJkQ · 2025-11-28
> >
> > Thank you for your response and for the revisions made to the paper. I have no further questions.

---

> > > ### Author Response · Authors · 2025-12-01
> > >
> > > Thank you so much for your positive evaluation and feedback! Your comments and suggestions are greatly valuable to our work!

---

### Official Review · Reviewer_pAAM · 2025-10-31

**Soundness:** 2
**Presentation:** 3
**Contribution:** 3
**Rating:** 6
**Confidence:** 4

**Summary:**

The paper targets a concrete missing piece in multi-track music generation: current models often produce plausible individual tracks that are not rhythmically aligned. To fix this, the authors propose SyncTrack, which (i) separates track-shared rhythm from track-specific instrument information, (ii) adds two cross-track attentions (global for overall groove, time-specific for exact same-time alignment), and (iii) introduces three rhythm-aware metrics (IRS, CBS, CBD) to evaluate stability and synchronization that FAD cannot capture. Experiments on 4-track Slakh2100 show better FAD and better rhythm metrics than recent multi-track baselines.

**Strengths:**

Very clear problem–solution alignment. The paper does not vaguely say “quality is low”; it pinpoints “rhythmic stability and cross-track synchronization are not modeled or evaluated,” and the proposed modules map 1:1 to that diagnosis.

Metrics with reuse potential. IRS/CBS/CBD are defined in a way that any multi-track model with separated stems can use; this makes the paper more than “a new model,” it is also “a more appropriate test.”

Realistic multi-track setup. Using four typical production tracks (bass, drums, guitar, piano) matches real downstream use (mixing, remixing, track-wise editing) much better than single-mix generation.

Empirical validation + robustness. The authors not only report scores but also check metric robustness w.r.t. beat-tracking hyperparameters and human subjective ratings, which strengthens the claim that the new metrics are meaningful.

Architecture that is easy to transplant. The separation between track-shared and track-specific modules is structurally simple and could be dropped into other latent-audio diffusion systems.

Conditional/partial generation not studied. A common real-world scenario is “I already have two tracks, generate a third one in sync.” The current setup is “generate all four together.” It would be good to see whether the same architecture can be conditioned on existing tracks and whether CBS/CBD still apply in that case.

**Weaknesses:**

Beat-detection dependency. All three rhythm metrics assume that beat/onset tracking works reasonably well. For highly expressive, weakly pulsed, or rubato multi-track music, tracking can fail, which would make IRS/CBS/CBD less reliable. The paper tests robustness to hyperparameters, but not to style changes; adding such a test would make the metric story stronger.

Dataset narrowness. Most results are on the Slakh2100 four-track configuration, which is clean and well-aligned. It is unclear how well SyncTrack and, more importantly, the metrics behave on genuine studio stems with small human timing jitter.

**Questions:**

Your three metrics all rely on a beat (or onset) extractor. How do you handle tracks that have very sparse or chordal content (e.g., piano pads) where the beat detector fails or produces inconsistent beat intervals? Do you ignore such tracks in IRS/CBS/CBD, or do you impute missing beats? What is the effect on CBS when one track is unreliable?

The two cross-track attention submodules are placed together in the track-shared module. Did you test alternative orderings (e.g., time-specific first, then global) or distributing them across different layers? If so, do results change significantly?

---

> ### Author Response · Authors · 2025-11-21
> **Response to Reviewer pAAM (1/2)**
>
> We are deeply grateful for your thoughtful and detailed review, which provides us with invaluable guidance for refining our approach and evaluation. Please find our point-by-point responses below.
>
> ---
>
> **S1: Conditional/partial generation not studied. A common real-world scenario is "I already have two tracks, generate a third one in sync." The current setup is "generate all four together." It would be good to see whether the same architecture can be conditioned on existing tracks and whether CBS/CBD still apply in that case.**
>
> >**Response:** Thank you for raising this important point. We have conducted additional experiments to validate that CBS/CBD metrics remain applicable in conditional generation scenarios. We test four configurations: (1) B: generation of Bass given the other three tracks; (2) BP: generation of Bass and Piano given the other two tracks; (3) BGP:  generation of Bass, Guitar, and Piano; (4) BDGP: generation of all four tracks together.
> >
> >The table below demonstrates that CBS and CBD metrics can be effectively computed across all these cases. Furthermore, the metric results align well with intuitive expectations regarding task difficulty. We would expect rhythm synchronization to be most challenging when generating all tracks (BDGP), followed by BGP, BP, and B (easiest). Our experimental results confirm this expected ordering in both CBS and CBD scores.
> >
> >We appreciate your feedback, which has helped us expand the application scenarios of CBS/CBD and further validate their rationality. We have added these experiments and corresponding discussions (**Section A.8 in the suuplementary**).
>
> |              | GT     | B   | BP  | BGP  | BDGP   |
> | ------------ | ------ | ------ | ------ | ------ | ------ |
> | CBS↑        | 0.5740 | 0.5620 | 0.5576 | 0.5259 | 0.5206 |
> | CBD(std)↓    | 0.1578 | 0.1902 | 0.1940 | 0.2118 | 0.2131 |
> | CBD(median)↓ | 0.2066 | 0.2192 | 0.2188 | 0.2304 | 0.2258 |
> | CBD(mean)↓    | 0.2412 | 0.2591 | 0.2579 | 0.2718 | 0.2681 |
>
> *Table 1: CBS and CBD metrics across different generation scenarios.*
>
> >**Modification in the manuscript**: Section A.8 in the supplementary.
>
> **W1: Beat-detection dependency. All three rhythm metrics assume that beat/onset tracking works reasonably well. For highly expressive, weakly pulsed, or rubato multi-track music, tracking can fail, which would make IRS/CBS/CBD less reliable. The paper tests robustness to hyperparameters, but not to style changes; adding such a test would make the metric story stronger.**
>
>
> >**Response:** We thank you for this insightful suggestion. To address robustness of our proposed metrics to style diversity, we have evaluated our proposed metrics on MUSDB18 [1], which encompasses various music styles including acoustic/folk pop, rhythm and blues/soul, etc. The table below compares metric scores between MUSDB18 and Slakh2100.
> >
> >As anticipated, MUSDB18 exhibits more diverse styles and consequently shows slightly weaker rhythm synchronization and stability compared to Slakh2100. This diversity is naturally reflected in our metrics. Nevertheless, both real datasets (MUSDB18 and Slakh2100) demonstrate superior coherence and stability compared to the generated music, particularly against MSDM and MSG-LD. These results further validate the effectiveness of our proposed metrics across different musical styles.
>
> >[1] Rafii Z, Liutkus A, Stöter F R, et al. The MUSDB18 corpus for music separation[J]. 2017.
>
> | Model/Dataset | CBS↑ | CBD(mean)↓ | CBD(std)↓ | CBD(median)↓ | IRS(B)↓ | IRS(D)↓ | IRS(G)↓ | IRS(P\)↓ |
> |---------------|------|------------|-----------|--------------|---------|---------|---------|---------|
> | MSG-LD        | 0.386 | 0.371      | 0.264     | 0.355        | 0.041   | 0.040   | 0.039   | 0.039   |
> | MSDM          | 0.469 | 0.313      | 0.222     | 0.281        | 0.050   | 0.036   | 0.034   | 0.046   |
> | SyncTrack     | 0.521 | 0.268      | 0.213     | 0.226        | 0.021   | 0.011   | 0.024   | 0.023   |
> | MUSDB18       | 0.512 | 0.264      | 0.197     | 0.223        | 0.017   | 0.007   | 0.029   | 0.015   |
> | Slakh2100     | 0.574 | 0.241      | 0.158     | 0.207        | 0.015   | 0.005   | 0.016   | 0.015   |
>
> *Table 2: Scores on three metrics for the MUSDB18 and Slakh2100.*
>
> > **Modification in the manuscript:** Section A.7 in the supplementary.

---

> ### Author Response · Authors · 2025-11-21
> **Response to Reviewer pAAM (2/2)**
>
> **W2: Dataset narrowness. Most results are on the Slakh2100 four-track configuration, which is clean and well-aligned. It is unclear how well SyncTrack and, more importantly, the metrics behave on genuine studio stems with small human timing jitter.**
>
> >**Response:** We appreciate this valuable comment. Building upon our response to W1, we further utilize MUSDB18 to evaluate our metrics on genuine studio data. The results we obtained are consistent with intuition and make sense.
> >
> >Specifically, MUSDB18 consists of genuine studio stems and features diverse musical styles, and exhibits more varied rhythm synchronization compared to the clean Slakh2100 dataset. Intuitively, rhythm stability and synchronization is worse than Slakh2100. As shown in Supplementary A.7, it can be observed that the distributions of the three proposed metrics on music from Slakh2100 are  better than those on MUSDB18. Despite this, MUSDB18 maintains overall better harmony and rhythmic stability than music generated by both MSG-LD and SyncTrack.
>
>
>
> >**Modification in the manuscript**: Section A.7 in the supplementary.
>
>
> **Q1: Your three metrics all rely on a beat (or onset) extractor. How do you handle tracks that have very sparse or chordal content (e.g., piano pads) where the beat detector fails or produces inconsistent beat intervals? Do you ignore such tracks in IRS/CBS/CBD, or do you impute missing beats? What is the effect on CBS when one track is unreliable?**
>
> > Thank you for these insightful questions. The scenarios you proposed can indeed help us better clarify the robustness of the metric.
> >
> >> **1.We don't ignore the tracks with sparse or chordal content.** Our metrics are designed to handle such cases directly and can effectively evaluate music quality across diverse scenarios.
> >>
> >> **2.When a track is unreliable due to sparse content, it does not compromise CBS's ability to reflect multi-track synchronization.** Speficically, we take two types of multi-track music as example: ⓐ four reliable tracks, and ⓑ three reliable tracks plus one sparse track. We next consider the following two cases:
> >>> (1) When all reliable tracks have beats that are very well aligned, the listener should not be able to distinguish any difference between ⓐ and ⓑ in terms of rhythm synchronization. In this case, the CBS scores for the two types of music are 4/4 and 3/3, respectively.
> >>>
> >>> (2) When both types have one track with unaligned beats, ⓑ should have a larger proportion of disharmony, making it more uncomfortable. In this case, the CBS scores are 3/4 and 2/3, which are more consistent with our intuition.
>
> **Q2: The two cross-track attention submodules are placed together in the track-shared module. Did you test alternative orderings (e.g., time-specific first, then global) or distributing them across different layers? If so, do results change significantly?**
>
> >**Response:** We appreciate your suggestions, which has helped us better justify SyncTrack's architectural design motivation. We conducted ablation studies with two alternative configurations: SyncTrack-reorder (swapping the order of global and time-specific attention) and SyncTrack-alternate (distributing the modules across different layers).
> >
> >The results demonstrate that our original SyncTrack configuration outperforms both variants significantly. This validates the importance of integrating both attention modules concurrently within each block. Moreover, the sequence of global attention followed by time-specific attention is well-founded, as it enables fine-grained synchronization building upon global rhythmic patterns.
> >
> | Model                        | Bass  | Drum  | Guitar | Piano | Mixture | Promotion |
> |------------------------------|-------|-------|--------|-------|---------|-----------|
> | SyncTrack-alternate          | 0.900 | 0.897 | 2.663  | 1.757 | 1.586   | 20.55%    |
> | SyncTrack-reorder            | 0.957 | 0.943 | 2.887  | 1.877 | 1.681   | 25.04%    |
> | **SyncTrack**                | **0.710** | **0.710** | **1.450**  | **1.110** | **1.260**   | -         |
>
> *Table 3: FAD of SyncTrack, SyncTrack-alternate and SyncTrack-reorder.*
> >**Modification in the manuscript**: Section 5.4 in the main text.
> ---
> Your insightful suggestions have greatly strengthened our paper, and we appreciate the time and expertise you dedicated to our work!

---

> > ### Comment · Reviewer_pAAM · 2025-11-26
> >
> > I have read the author's response and other reviewers' opinions. Currently, I have no further questions.

---

> > > ### Author Response · Authors · 2025-11-27
> > >
> > > Thank you so much once again for taking the time to help us improve the paper. We are glad to your positive feedback! Your feedback is invaluable to the creation of a better version of our manuscript. Thanks!

---

### Official Review · Reviewer_FeMV · 2025-11-02

**Soundness:** 3
**Presentation:** 2
**Contribution:** 3
**Rating:** 6
**Confidence:** 4

**Summary:**

This paper introduces SyncTrack, a new model for synchronous multi-track waveform music generation. The model uses an innovative architecture to tackle the often-overlooked challenges of rhythmic stability and synchronization in multi-track music. It also incorporates two types of cross-track attention mechanisms: global cross-track attention, which ensures the rhythm stays consistent across tracks, and time-specific cross-track attention, which helps align musical events more precisely in time. The authors conducted thorough experiments to demonstrate the effectiveness of their approach, showing that it performs well in creating high-quality multi-track music.

**Strengths:**

1.The modules designed by the authors are highly aligned with the current needs of music generation technologies, and their effectiveness is demonstrated from both temporal and spatial perspectives.

2.The authors propose three novel metrics to evaluate the quality of multi-track music generation, addressing gaps in existing evaluation methods.

3.The experiments conducted by the authors are thorough and provide strong evidence supporting the effectiveness of the proposed approach.

**Weaknesses:**

1.The authors do not provide a clear explanation of the relationship between tracks, timbre, and rhythm in the music generation task, which may cause some difficulty in understanding the proposed approach.

2.The ablation studies are limited and the results are relatively average, making them less convincing in demonstrating the contributions of specific components.

3.The paper does not include a user study, leaving the subjective evaluation of the generated music quality unaddressed.

**Questions:**

1.Besides the FAD metric and the three proposed in the paper, are there other commonly used metrics for evaluating the quality of generated audio?

2.Why didn't the authors include a user study to subjectively score the generated results?

3.What is the relationship between different tracks and aspects such as rhythm and timbre? Does the model impact the quality of individual track generation?

---

> ### Author Response · Authors · 2025-11-21
> **Response to Reviewer FeMV (1/3)**
>
> We sincerely appreciate your constructive feedback and insightful comments on our work! We have carefully considered all the points you raised and have revised the manuscript accordingly. Our point-by-point responses are detailed below.
>
> ---
>
> **W1 \& Q3: The authors do not provide a clear explanation of the relationship between tracks, timbre, and rhythm in the music generation task, which may cause some difficulty in understanding the proposed approach. Does the model impact the quality of individual track generation?**
>
> >**Response:** Thank you for this valuable feedback, as it also gives us the opportunity to clarify our motivations.
> >
> >**1. Relationship between tracks, timbre, and rhythm**. In multi-track music generation, the objective is to generate multiple instrumental layers (e.g., bass, drums), where each track remains independent yet coherently integrated. Two core aspects govern this process: rhythm and timbre.
> >
> >> (1) **Rhythm** refers to the structured temporal arrangement of musical events. **Within a single track**,  rhythm must exhibit **stability**, meaning that note onsets and beats adhere consistently to a regular metrical grid. **Across multi tracks**, rhythm requires **synchronization**, ensuring that events from different instruments align precisely with each other and a shared underlying pulse.
> >>
> >> (2) **Timbre** defines the unique perceptual quality of sound that distinguishes one instrument from another, such as the warm character of a bass or the bright attack of a piano.
> >
> >When constructing our model, we designed three dedicated modules: (1) track-specific module, (2) global cross-track attention mechanism, and (3)time-specific cross-track attention mechanism, to explicitly capture information related to timbre, rhythm stability, and rhythm synchronization, respectively.
>
> >**Modification in the manuscript**: Introduction (Paragraph II) in the main text.
>
>
> >**2. Impact of SyncTrack on the quality of individual track generation.** Our model does improve individual track generation:
> >>(1) **Objectively**, as shown in the following table (**also summarized in Table 1 and Table 2 in the maintext**), SyncTrack reduces both IRS and FAD scores for all individual tracks, indicating better track-wise quality and rhythmic stability.
> >>(2) **Subjectively**, we conduct a listening test to evaluate the quality of individual tracks. Participants were presented with single-track audio generated by SyncTrack, MSG-LD, and the Ground Truth from the dataset. The results showed that SyncTrack received an average subjective score of 1.86, outperforming MSG-LD (1.16) and approaching the ground truth (2.74) (**also summarized in  Table 3 in the maintext**).
> >>(3) **Intuitively**, temporal prediction errors caused by irregular rhythms or inter-track misalignment lead to persistent auditory discomfort [1]. Since our model generates music with superior rhythmic stability and synchronization, it inherently reduces these artifacts, resulting in a more pleasant listening experience and higher perceived quality for each individual track.
>
>
> | Method    | Bass      |           | Drum      |           | Guitar    |           | Piano     |           |
> |-----------|:---------:|:---------:|:---------:|:---------:|:---------:|:---------:|:---------:|:---------:|
> | **Metrics** | **IRS↓** | **FAD↓** | **IRS↓** | **FAD↓** | **IRS↓** | **FAD↓** | **IRS↓** | **FAD↓** |
> | SyncTrack | **0.021** | **0.710** | **0.011** | **0.710** | **0.024** | **1.450** | **0.023** | **1.110** |
> | MSG-LD    | 0.041     | 1.050     | 0.040     | 0.980     | 0.039     | 1.830     | 0.039     | 2.040     |
>
> *Table 1: Quality of individual track generation (SyncTrack and MSG-LD）*
>
> >[1] Ernest Mas-Herrero, Alain Dagher, and Robert J Zatorre. Modulating musical reward sensitivity up and down with transcranial magnetic stimulation. Nature human behaviour, 2(1):27-32, 2018.
>
> >**Modification in the manuscript:** Table 1, 2, 3 in the main text.

---

> ### Author Response · Authors · 2025-11-21
> **Response to Reviewer FeMV (2/3)**
>
> **W2: The ablation studies are limited and the results are relatively average, making them less convincing in demonstrating the contributions of specific components.**
>
> >**Response:** Thank you for this valuable suggestion.
> >
> >>**1. To address the issue of the limited ablation study**, we have redesigned our ablation study setup. SyncTrack is built upon the Backbone model by incorporating three key modules: ⓐ track-specific module, ⓑ global cross-track attention, ⓒ time-specific cross-track attention. Note that ⓑ and ⓒ appear sequentially within each track-shared module. We consider six ablation variants here: (1) Backbone; (2) Backbone w/ ⓐ; (3) Backbone w/ ⓐ+ⓑ; (4) Backbone w/ ⓐ+ⓒ; (5) SyncTrack-reorder, which reverses the order of these two attention modules ⓑ and ⓒ; (6) SyncTrack-alternate, which uses only one of ⓑ or ⓒ in each track-shared module alternatively.
> >>
> >>As shown in the following table,  we report FAD scores for all models and the FAD improvement (Promotion) of SyncTrack’s final mixed music over its ablated variants. Our experiments address the following questions:
> >>>(1) **Q**: Are all three modules useful?
> >>>
> >>>**A:** Yes. SyncTrack achieves significant improvement over the backbone (50\%) and variants (11\%-27\%).
> >>>
> >>>(2) **Q**: Do the three modules play distinct roles?
> >>>
> >>>**A:** The ablation studies confirm that each of the three modules serves a distinct and complementary function. 1) ⓐ captures timbre and other track-specific features, resulting in a significant improvement in individual-track audio quality (ranging from 56.18% to 84.41%). 2) ⓑ enhances the stability of each track. Hence, Backbone w/ ⓐ+ⓑ has significant improvement (6.3\%-22.55\%) over Backbone w/ ⓐ in single-track music quality. 3) ⓒ enables fine-grained synchronization, aligning musical events across tracks at the same temporal position. Backbone w/ ⓐ+ⓒ shows significantly enhances in the overall quality (17.97%) of multi-track music over the backbone w/ ⓐ.
> >>>
> >>>(3) **Q:** Is the integration of modules well-reasoned?
> >>>
> >>>**A:** Yes. SyncTrack outperforms both SyncTrack-reorder and SyncTrack-alternate variants, supporting the design choice of incorporating both ⓑ and ⓒ concurrently as in the proposed order. Since ⓒ enables a more fine-grained synchronization building upon ⓑ, the sequence of first ⓑ followed by ⓒ is well-founded.
>
>
> >>**2. Regarding the concern on relatively average results**, we report the FAD improvement on the final mixed music achieved by SyncTrack compared to its variants. we note that SyncTrack shows significant and consistent improvements (>10% mixture FAD) compared to all variants.
>
> | Model                        | Bass  | Drum  | Guitar | Piano | Mixture | Promotion |
> |------------------------------|-------|-------|--------|-------|---------|-----------|
> | Backbone                     | 5.234 | 3.081 | 6.012  | 6.170 | 2.570   | 50.97%    |
> | Backbone w/ ⓐ              | 0.816 | 0.809 | 2.634  | 1.695 | 1.742   | 27.67%    |
> | Backbone w/ ⓐ+ⓑ          | *0.632* | *0.758* | *2.367*  | *1.359* | 1.627   | 22.56%    |
> | Backbone w/ ⓐ+ⓒ          | 0.892 | 0.889 | 2.680  | 1.547 | *1.429*   | 11.83%    |
> | SyncTrack-alternate          | 0.900 | 0.897 | 2.663  | 1.757 | 1.586   | 20.55%    |
> | SyncTrack-reorder            | 0.957 | 0.943 | 2.887  | 1.877 | 1.681   | 25.04%    |
> | **SyncTrack**                | 0.710 | **0.710** | **1.450**  | **1.110** | **1.260**   | -         |
>
> *Table 2: FAD of SyncTrack and six ablation variants. Italics indicate the best performance among all variants.*
>
> >**Modification in the manuscript:** Section 5.4 and Table 6 in the main text.

---

> ### Author Response · Authors · 2025-11-21
> **Response to Reviewer FeMV (3/3)**
>
> **W3 & Q2: The paper does not include a user study, leaving the subjective evaluation of the generated music quality unaddressed.**
>
> >**Response:** Thanks for your kindly suggestion, which make our evaluation of the music quality generated by SyncTrack more detailed and complete. We have now incorporated the subjective evaluation results (**previously in the supplementary material, Section A.6**) into the main text (**Section 5.1**). And as you suggested, a detailed summary of subjective ratings is now added in the tabular form (**Table 3 in the main text**).
>
> >**Modification in the manuscript:** Section 5.1 and Table 3 in the main text
>
> **Q1: Besides the FAD metric and the three proposed in the paper, are there other commonly used metrics for evaluating the quality of generated audio?**
>
> >**Response:** Thank you for this question. Yes, there are other metrics such as KL divergence and Fréchet distance, which we also compute using the `audioldm_eval`$^1$ tool. For example: (1) KL divergence($\uparrow$): MSG-LD = 0.4193, SyncTrack = 0.4756; (2) Fréchet distance($\downarrow$): MSG-LD = 2.8380, SyncTrack = 2.300
> >
> > However, these metrics also reflect the similarity between the generated music and the reference dataset (Slakh100 test set), which is quite similar to FAD. For this reason, both MSDM [1] and MSG-LD [2] only use FAD for testing. Thus, we followed the same convention to ensure a fair and consistent comparison and do not include these metrics in the paper either.
>
> > [1] Mariani G, Tallini I, Postolache E, et al. Multi-Source Diffusion Models for Simultaneous Music Generation and Separation[C]//ICLR. 2024.
> >
> > [2] Karchkhadze T, Izadi M R, Dubnov S. Simultaneous music separation and generation using multi-track latent diffusion models[C]//ICASSP 2025-2025 IEEE International Conference on Acoustics, Speech and Signal Processing (ICASSP). IEEE, 2025: 1-5.
> >
> >1.https://github.com/haoheliu/audioldm_eval
> >
> ---
> Thank you once again for your valuable comments and for helping us enhance the clarity and quality of our work.

---

> > ### Comment · Reviewer_FeMV · 2025-11-27
> > **Replying to Rebuttal**
> >
> > Thank you for your response. I have no further questions. The authors have supplemented and revised the original text based on my concerns and suggestions, addressing some of my doubts in terms of presentation. The model design in this paper is effective, but its overall practicality in the field is limited. Therefore, I maintain my score.

---

> ### Author Response · Authors · 2025-11-29
>
> Thank you very much for your careful re-examination of our revised manuscript and for your constructive comments. We truly appreciate your positive assessment that “the model design in this paper is effective.” In response to your remaining concern regarding the overall practicality of our approach in real-world scenarios, we would like to provide further clarification.
>
> Recently, the field of music generation has seen growing interest in multi-track generation, which enables flexible editing and practical applications—such as track mixing, rearrangement, and the addition of new instruments. Within this context, our work tackles the long-overlooked issue of rhythmic stability and synchronization in multi-track music. To address this, we design **SyncTrack**, a model that incorporates both **track-shared modules** and **track-specific modules**, enabling it to jointly **capture rhythmic synchronization across tracks while preserving track-specific timbral and individual characteristics**.
>
> Beyond multi-track music generation, our approach is also applicable to a variety of other domains that require synchronization. **As Reviewer pAAM noted, “the track-shared and track-specific modules are structurally simple and can be easily dropped into other latent-audio diffusion systems.”** This **plug-and-play** design allows for broad applicability. For instance, tasks such as accompaniment generation [1] (generating accompaniments from a melody), singing voice generation [2,3] (synthesizing vocals from instrumental tracks), and full song generation [4] (separately generating both vocals and melodies) all place a strong emphasis on rhythmic synchronization. These models all use a diffusion architecture, which **allows our modules to be seamlessly integrated into them**. This is also the work we plan to do next.
>
> Moreover, **conditional or partial generation** is a common real-world scenario. For instance, a user might have two existing tracks and wish to generate a third, synchronized one. This scenario also benefits directly from our method. Our modules can be readily integrated into such systems, and we have empirically validated their effectiveness in conditional generation settings. Results are summarized in the following table.
>
>
> |              | GT     | B   | BP  | BGP  | BDGP   |
> | ------------ | ------ | ------ | ------ | ------ | ------ |
> | CBS↑        | 0.5740 | 0.5620 | 0.5576 | 0.5259 | 0.5206 |
> | CBD(std)↓    | 0.1578 | 0.1902 | 0.1940 | 0.2118 | 0.2131 |
> | CBD(median)↓ | 0.2066 | 0.2192 | 0.2188 | 0.2304 | 0.2258 |
> | CBD(mean)↓    | 0.2412 | 0.2591 | 0.2579 | 0.2718 | 0.2681 |
>
>
>
> To evaluate rhythmic quality, we also introduce new metrics, which represent another practical contribution of our work. Compared to conventional metrics such as FAD, our proposed metrics offer two major advantages:
>
> 1) They are **specifically designed to assess rhythmic aspects of music**, are applicable across diverse scenarios, and have been validated to produce meaningful and interpretable results.
>
> 2) They enable **per-sample quality evaluation**, whereas FAD only measures aggregate performance over large batches. This fine-grained assessment also supports practical use cases, such as allowing users to quickly filter and select high-quality music segments.
>
> We would like to once again express our sincere gratitude for your valuable feedback. We are glad that you acknowledged our efforts to improve the clarity of the manuscript and address several of your earlier concerns. We hope the revised version is now clearer and more coherent.
>
>
> [1] Donahue C, Caillon A, Roberts A, et al. SingSong: generating musical accompaniments from singing (2023)[J]. URL https://arxiv. org/abs/2301.12662, 2023.
>
> [2] Sui K, Xiang J, Jin F. SmoothSinger: A Conditional Diffusion Model for Singing Voice Synthesis with Multi-Resolution Architecture[J]. arXiv preprint arXiv:2506.21478, 2025.
>
> [3] Guo W, Zhang Y, Pan C, et al. Techsinger: Technique controllable multilingual singing voice synthesis via flow matching[C]//Proceedings of the AAAI Conference on Artificial Intelligence. 2025, 39(22): 23978-23986.
>
> [4] Ning Z, Chen H, Jiang Y, et al. DiffRhythm: Blazingly fast and embarrassingly simple end-to-end full-length song generation with latent diffusion[J]. arXiv preprint arXiv:2503.01183, 2025.

---

### Author Response · Authors · 2025-11-27
**General Response (1/2)**

**Dear Reviewers, Area Chairs, Senior Area Chairs, and Program Chairs**,

We sincerely thank all reviewers for their positive and constructive feedback. The reviewers highlighted several key strengths of our work, including **the novelty of our methodology, the practical relevance of our evaluation setup, the thoroughness of our experimental validation, and the clarity and modularity of our implementation**. We are especially encouraged that the reviewers recognized the **significance of our proposed evaluation metrics and the broader applicability of our multi-track generation framework**.

Below, we summarize the key strengths of our paper as identified by the reviewers:

**[Novelty in Methodology & Real-World Relevance]:**
- **Reviewer FeMV**: "The modules designed by the authors are highly aligned with the current needs of music generation technologies."
- **Reviewer pAAM**: "Very clear problem–solution alignment."
- **Reviewer 1JZQ**: "Based on a comprehensive evaluation by FAD and a focused assessment by IRS, the proposed idea appears to successfully achieve its goal."


**[Impactful Evaluation & Performance]:**
- **Reviewer FeMV**: "The authors propose three novel metrics to evaluate the quality of multi-track music generation, addressing gaps in existing evaluation methods."
- **Reviewer pAAM**: "The realistic multi-track setup using four typical production tracks better reflects real downstream use compared to single-mix generation."
- **Reviewer CJkQ**: "The paper presents clear and reproducible rhythm-centric evaluation metrics (IRS/CBS/CBD) with robustness analyses, enhancing their potential for broader community adoption."

**[Comprehensive Validation & Analysis]:**
- **Reviewer FeMV**:  "The experiments conducted by the authors are thorough and provide strong evidence supporting the effectiveness of the proposed approach."
- **Reviewer pAAM**: "The authors not only report scores but also check metric robustness."
- **Reviewer CJkQ**: "The results demonstrate good alignment between objective and subjective evaluations."
- **Reviewer 1JZQ**: "The paper also presents additional experiment results with ablation and subjective tests."

**[Portability & Practical Implementation]**:
- **Reviewer pAAM**: "Architecture that is easy to transplant. The separation between track-shared and track-specific modules is structurally simple and could be dropped into other latent-audio diffusion systems."


To address the reviewers' constructive comments, we have undertaken comprehensive **revisions** focusing on four key areas:

 (1) **A thorough analysis of the proposed metrics' practical applicability and generalizability across diverse real-world scenarios**;
 (2) **An expanded ablation study that systematically validates the design and integration of SyncTrack's core modules**;
 (3) **A significantly elaborated user study with full protocols and detailed results to enhance reproducibility and interpretability**;
 (4) **The addition of critical background, efficiency analysis, model specifications, and code to improve clarity and reproducibility.**

We have now conducted additional experiments, clarified points, and engaged in discussions to address the valuable comments provided by the reviewers. Based on the constructive feedback, we have carefully *revised the manuscript of our work using blue highlights*.

---

> ### Author Response · Authors · 2025-11-27
> **General Response (2/2)**
>
> Specifically,
> - For **(1)**, we extend our experiments to demonstrate that our metrics can be applied to music of different quality levels, styles, segment lengths, beat-detection configurations, and music generation tasks.
>
> | New results | Application scenarios |  Main text| Supplementary|Response to|
> |-|-|-|-|-|
> | -        | Well-aligned music dataset |    Table 4,5  |Figure A3, A4|-|
> | -        | Beat tracking configurations |  -  |Figure A1, A2, Table A1|-|
> | ✓       |Genuine studio recordings|-|Table A4, Figure A7|Reviewer pAAM (W2) & Reviewer 1JZQ (W3)|
> | ✓       |Different music style|-|Table A4, Figure A7|Reviewer pAAM W1 & Reviewer 1JZQ (W3)|
> |✓|Different music segments|-|Table A6, Figure A8| Reviewer CJkQ W1 & Reviewer 1JZQ (W3)|
> |✓|Different music generation task|-|Table A5|Reviewer pAAM （s1) & Reviewer 1JZQ (W3)|
>
>
>
> - For **(2)**, SyncTrack builds upon the Backbone model by incorporating three key modules: ⓐ track-specific module, ⓑ global cross-track attention, ⓒ time-specific cross-track attention. Our experiments address the following three questions: **Q1:** Are all three modules useful? **Q2:** Do the three modules play distinct roles? **Q3:** Is the integration of modules well-reasoned?
>
> | New results | Model |  Goal|Question|Response to|
> |-|-|-|-|-|
> | -        | Backbone |    -| Q1 & Q2|-|
> | -        | Backbone w/ ⓐ | Verify ⓐ compard to Backbone | Q2|-|
> | -       |Backbone w/ ⓐ+ⓑ|Verify ⓑ compard to Backbone w/ ⓐ| Q2|-|
> | -       |Backbone w/ ⓐ+ⓒ |Verify ⓒ compard to Backbone w/ ⓐ|Q2|-|
> |✓|SyncTrack-alternate|Verify the num of ⓑ & ⓒ compared to SyncTrack |Q3|Reviewer pAAM (Q2) & Reviewer FeMV (W2)|
> |✓|SyncTrack-reorder|Verify the order of ⓑ & ⓒ compared to SyncTrack|Q3|Reviewer pAAM (Q2) & Reviewer FeMV (W2)|
> |-|SyncTrack|-|     Q1 & Q2 & Q3|-|
>
>
>
>
> - For **(3)**, to enhance reproducibility, we have provided more details of user study protocol and the complete per-participant scoring data.
>
> | New | Details |  Main text| Supplementary|Response to|
> |-|-|-|-|-|
> | Modified        |  User study  protocol |    Section 5.1  |Section A.6|Reviewer CJkQ (W2&Q1) & Reviewer FeMV (W3&Q2)|
> | -        | Correspondence between objective metric and subjective scores |  Figure 4  |-|-|
> | ✓       |Subjective evaluation results|Table 3|-|Reviewer CJkQ (W2&Q1)|
>
>
>
> - For **(4)**, we expand the background section (Reviewer 1JZQ W1), provide a detailed efficiency and time analysis (Reviewer CJkQ Q2), and add the detailed model specifications ( Reviewer 1JZQ W2&Q1). Additionally, to ensure full reproducibility, we have provided a link to our anonymous code.
>
> We look forward to the reviewers' feedback on these comprehensive updates.
>
> Best regards,
>
> Paper 1918 Authors

---

### Meta-Review · Area_Chair_jznu · 2026-01-07

**Summary:**

The initial ratings are 6, 6, 6, 4. This paper proposes a model for synchronous multi-track waveform music generation.It incorporates two cross-track attention mechanisms: global cross-track attention, which ensures the rhythm stays consistent across tracks, and time-specific cross-track attention, which helps align musical events more precisely in time. The authors conducted thorough experiments to demonstrate the effectiveness of their approach.

Strengths:
(1)The paper introduces clear and reproducible rhythm-centric evaluation metrics (IRS/CBS/CBD) and provides robustness analyses, making these metrics potentially useful for broader community adoption.
(2)The results demonstrate good alignment between objective and subjective evaluations: FAD improvements are consistent with listening preferences, and component-level ablations further validate the architectural design.

Weaknesses:
(1) The discription of some technology details are lacked.
(2)The current setup (10.24 s at 16 kHz) constrains both long-range musical structure and high-frequency detail. Including evaluations on longer segments (≥30–60 s) or full-song contexts would strengthen the claims.

**Reviewer Concerns:**

Most concerns of Reviewer FeMV and CJkQ were addressed by the rebuttal, and some concerns of  Reviewer pAAM and 1JZQ are still outstanding.

**Reviewer Scores:**

Reviewer FeMV and CJkQ maybe raise the score.

---

### Decision · Program_Chairs · 2026-01-26

Accept (Poster)